# TRPγ regulates lipid metabolism through *Dh44* neuroendocrine cells

**Dharmendra Kumar Nath†, Subash Dhakal†, Youngseok Lee***

Department of Bio and Fermentation Convergence Technology, Kookmin University, Seoul, Republic of Korea

## eLife Assessment

This **important** study reports findings that Trpγ, a type of transient receptor potential (TRP) channel expressed in Dh44-releasing neuroendocrine cells, mediates starvation-dependent lipid catabolism. Overall, the claims of the authors are supported by **solid** evidence. The work should be of interest to both basic and medical biologists working on lipid metabolism.

**\*For correspondence:**
ylee@kookmin.ac.kr

†These authors contributed equally to this work

**Competing interest:** The authors declare that no competing interests exist.

## Abstract

Understanding how the brain controls nutrient storage is pivotal. Transient receptor potential (TRP) channels are conserved from insects to humans. They serve in detecting environmental shifts and in acting as internal sensors. Previously, we demonstrated the role of TRPγ in nutrient-sensing behavior (Dhakal et al., 2022). Here, we found that a TRPγ mutant exhibited in *Drosophila melanogaster* is required for maintaining normal lipid and protein levels. In animals, lipogenesis and lipolysis control lipid levels in response to food availability. Lipids are mostly stored as triacylglycerol in the fat bodies (FBs) of *D. melanogaster*. Interestingly, *trpγ* deficient mutants exhibited elevated TAG levels and our genetic data indicated that *Dh44* neurons are indispensable for normal lipid storage but not protein storage. The *trpγ* mutants also exhibited reduced starvation resistance, which was attributed to insufficient lipolysis in the FBs. This could be mitigated by administering lipase or metformin orally, indicating a potential treatment pathway. Gene expression analysis indicated that *trpγ* knockout downregulated *brummer*, a key lipolytic gene, resulting in chronic lipolytic deficits in the gut and other fat tissues. The study also highlighted the role of specific proteins, including neuropeptide DH44 and its receptor DH44R2 in lipid regulation. Our findings provide insight into the broader question of how the brain and gut regulate nutrient storage.

## Introduction

Nutrient storage is not a simple matter of accumulation but involves a sophisticated network of signals and receptors, feedback loops, and hormonal influences, all choreographed by the brain. The mechanisms involve various tissues and organs, including the liver, muscles, and adipose tissue, each playing a distinct role in the storage and release of vital nutrients like glucose, fats, and proteins in mammals (*Efeyan et al., 2015*). Key to this regulation are specialized cells and molecular pathways that respond to dietary intake, energy expenditure, and changes in the internal and external environment (*Tran et al., 2022*). For instance, the brain receives and interprets signals about the body's energy status through hormones such as insulin and leptin, and nutrients themselves, which inform decisions on whether to store or mobilize energy reserves in mammals (*Jais and Brüning, 2022*; *Roh and Kim, 2016*). Moreover, the role of TRP channels as molecular sensors that detect changes in the environment and internal metabolic status highlights the level of molecular sophistication involved in nutrient storage and energy balance (*Clapham, 2003*; *Dhakal and Lee, 2019*; *Fowler and Montell, 2013*; *Turner et al., 2016*). These channels bridge external stimuli with internal responses, ensuring

the body adapts efficiently to maintain homeostasis (*Dhakal and Lee, 2019*; *Venkatachalam and Montell, 2007*). This concert of mechanisms, from the molecular to the systemic level, exemplifies the elegance with which the brain ensures our energy needs are met in a changing environment, highlighting the intricate interplay of biological process that sustains life (*Picard et al., 2018*). Understanding these process opens up new vistas in our approach to diet, disease management, and overall health optimization.

The use of a genetically tractable model organism such as *Drosophila melanogaster* could provide fundamental insights into the potential relationship between TRP channels and metabolic syndrome. This study delves into the function of TRPγ channels, specifically their role in managing lipid and protein levels. Through genetic manipulation, we explored the consequences of TRPγ mutations on the metabolic processes, revealing critical insights into the regulation of lipid storage and breakdown. We found that TRPγ expression in *Dh44* neuroendocrine cells in the brain is critical for maintaining normal carbohydrate levels in tissues (*Dhakal et al., 2022*). Building on this, we hypothesized that TRPγ in *Dh44* cells also regulates lipid and protein homeostasis. Our investigation sheds light on the balance between lipogenesis and lipolysis, processes that, respectively, synthesize and degrade lipids based on the availability of food, and how these processes are affected by TRPγ activity. By examining the metabolic anomalies in *trpγ* deficient mutants, including alterations in lipid storage and the consequent impact on starvation resistance, this study advances our understanding of metabolic regulation. Furthermore, we uncover the potential therapeutic benefits of lipase and metformin in counteracting the metabolic deficiencies caused by TRPγ mutations. Through detailed gene expression analysis, we identify the downregulation of crucial genes involved in lipid metabolism, offering new perspectives on the genetic underpinnings of nutrient storage regulation. The implications of this research are far-reaching, not only enhancing our comprehension of basic biological functions but also providing avenues for therapeutic interventions in metabolic disorders.

## Results

### Trpγ mutants exhibit reduced sugar levels alongside elevated lipid and protein levels

In order to understand how *D. melanogaster* regulates the maintenance of major nutrient levels, including carbohydrates, lipids, and proteins, TRP channel mutants were examined as potential candidates. A recent study revealed that among TRP channel mutants, only *trpγ* mutant displayed only reduced carbohydrate levels (*Dhakal et al., 2022*), suggesting the involvement of additional TRP channels in regulating major nutrients. *D. melanogaster* possesses 13 members of the TRP channel family. Flies with mutations in *trpM*, *trpML*, and *nompC* showed high mortality rates with minimal survival, whereas those with mutations in the remaining 10 TRP superfamily genes were viable and healthy when homozygous. Investigation into the lipid and protein levels of other available TRP superfamily mutants under sated conditions revealed that, apart from the *trpγ* mutant, all tested mutants exhibited normal levels (*Figure 1A and B*). Specifically, the *trpγ¹* mutant showed 1.3 times higher triacylglycerol (TAG) levels and 1.5 times higher protein levels compared to the control. These metabolic changes were unique and specific to the *trpγ* mutants among the evaluated TRP superfamily mutants. Previous observations indicated that *trpγ* mutants had lower cellular sugar levels and stored glycogen (*Dhakal et al., 2022*). Consistently, *trpγ* mutants exhibited reduced levels of cellular sugars (glucose and trehalose), stored glycogen, and sugar levels in the hemolymph (*Figure 1C–E*). In conclusion, it was found that *trpγ¹* mutants displayed lower carbohydrate levels but higher lipid and protein levels compared to the control. Importantly, the increased TAG and protein levels observed in *trpγ¹* mutant were confirmed with a second *trpγ¹* allele (*trpγ^{G4}*) (*Figure 1F and G*). Furthermore, we restored the deficiency in lipid and protein levels using a genomic *trpγ⁺* transgene, *g(trpγ)*, and by introducing a *UAS-trpγ* cDNA, controlled by *GAL4* knocked into the *trpγ* locus (*trpγ^{G4}* flies; +).

### Trpγ regulates lipid metabolism through dh44 neuroendocrine cells in the pars intercerebralis

We recently proposed that TRPγ expression in the six DH44 neuroendocrine cells in the pars intercerebralis (PI) located in the dorsal medial area of the brain (*Figure 2—figure supplement 1A1–A3*) is essential to maintain normal carbohydrate levels in tissues (*Dhakal et al., 2022*). Therefore, we next

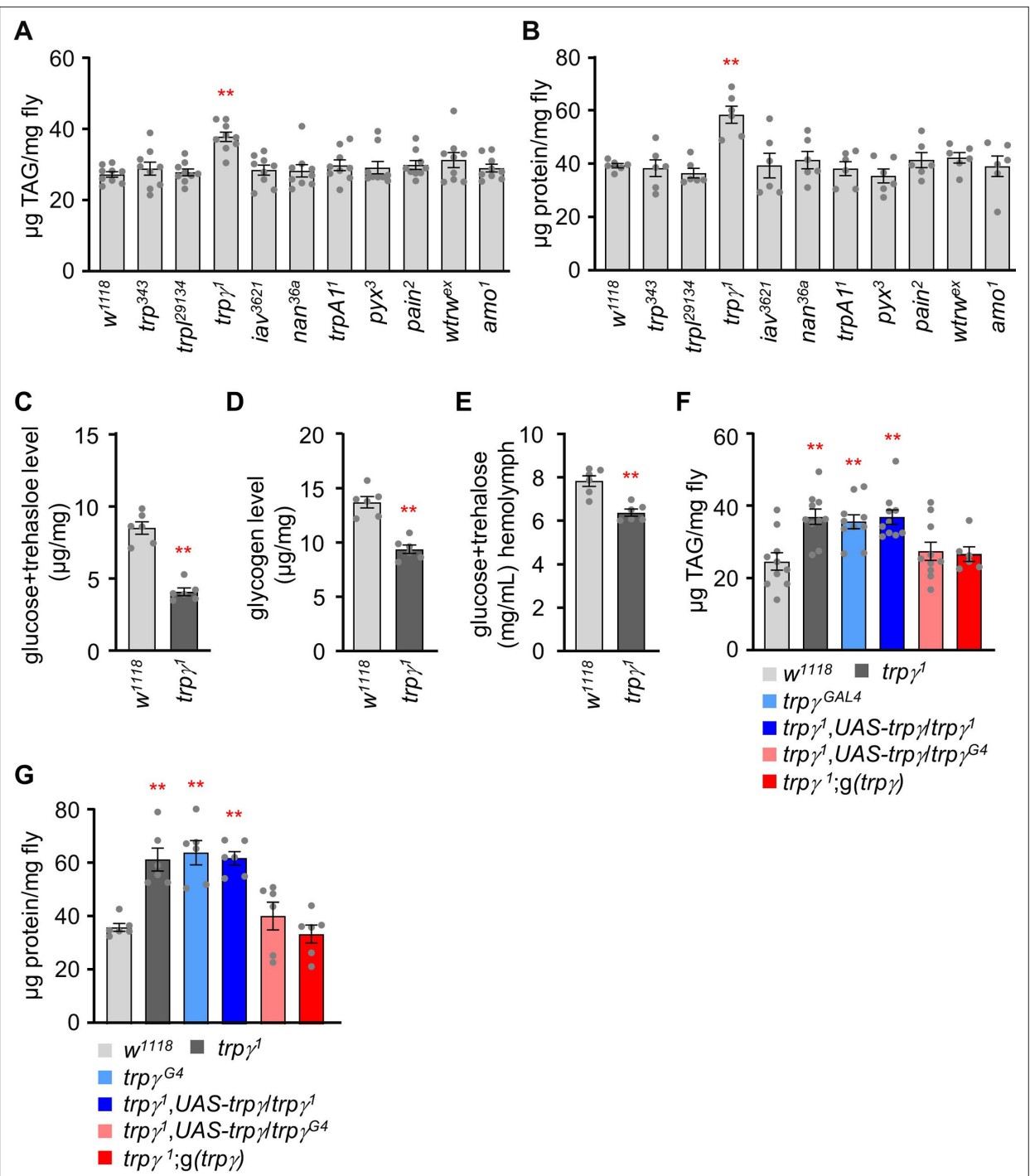

**Figure 1.** TRPγ mutants exhibit altered carbohydrate, lipid, and protein levels. (**A**) Triacylglycerol (TAG) level measurement in the whole-body extract from control ($w^{1118}$) and transient receptor potential (TRP) channel mutant lines (n=9). (**B**) Protein level measurement in the whole-body extract from control ($w^{1118}$) and TRP channel mutant lines (n=6). (**C**) Measurement of total glucose and trehalose levels (μg/mg) in the whole-body extracts of control ($w^{1118}$) and $trpγ^1$ adult males (n=6). (**D**) Measurement of tissue glycogen levels (μg/mg) in adult control ($w^{1118}$) and $trpγ^1$ males (n=6). (**E**) Measurement of Hemolymph glucose +trehalose level in the male flies of control ($w^{1118}$) and $trpγ^1$ (n=6). (**F**) Measurement of TAG level in adult males to test for rescue of the TAG defect in $trpγ$ flies with the $UAS$-$trpγ$ and the $trpγ^{G4}$ or with the $trpγ$ genomic transgene (n=6–10). (**G**) Rescue experiments showing the measurement of protein levels in adult flies with the indicated genotypes (n=6). Comparisons between multiple experimental groups were conducted via single-factor ANOVA coupled with Scheffe's *post hoc* test. The asterisks indicate significant differences from the controls (**p<0.01).

sought to determine whether protein and lipid levels can be regulated by the same neuroendocrine cells. To test this, we used a transgenic fly encoding an inwardly rectifying K$^+$ channel (*UAS-Kir2.1*) to inactivate specific neurons (*Hodge, 2009*). Interestingly, inactivating the *Dh44* neurons significantly increased the lipid levels but not protein levels (*Figure 2A and B*). Approximately 16 neurons in the PI express dILP2, which are also overlapped by two DH44-positive neurons in 5–10- d-old adult male flies (*Figure 2—figure supplement 1B1–B3*; *Ohhara et al., 2018*). However, inactivation of *dILP2* neurons did not impair either lipid or protein levels (*Figure 2A and B*). In conclusion, our findings indicated that DH44 neuroendocrine cells contribute to lipid regulation but not protein regulation.

To assess whether *trpγ* expression in *Dh44* neurons is sufficient to restore normal lipid and protein levels in *trpγ* mutant flies, we expressed the *UAS-trpγ* under the control of *Dh44-GAL4* in the *trpγ¹* mutant background (*Figure 2C and D*). Our findings indicated that the increased lipid levels in the *trpγ¹* mutant background decreased to normal levels via the expression of the *trpγ* transgene in the DH44 neuroendocrine cells but not in its parent strains (*Figure 2C*). Again, the expression of the *trpγ* transgene in the dILP2 neurons had no appreciable effects (*Figure 2C*). In contrast, the expression of the *trpγ* transgene did not decrease the protein levels in either the DH44 or the dILP2 neurons (*Figure 2D*). This indicates that TRPγ expressed in *Dh44* cells is sufficient for the regulation of lipid levels. Next, RNAi-mediated knockdown experiments were conducted to further examine the role of *trpγ* in *Dh44* neurons. Interestingly, *trpγ* knockdown in *Dh44* neurons significantly increased lipid levels, whereas flies harboring only the *Dh44-GAL4* or the *UAS-trpγ^RNAi* transgenes displayed normal lipid levels (*Figure 2E*). This indicated that TRPγ in *Dh44* cells is needed for the regulation of lipid levels. Furthermore, the total TAG level in the *trpγ¹* flies was higher in both males and females (*Figure 2— figure supplement 1C*), meaning that the functions of *trpγ* are not sex-specific.

In *D. melanogaster*, lipids are mainly stored in the form of TAG and cholesterol ester in the adipose tissue [i.e., fat bodies (FBs)] as lipid droplets (LDs) (*Liu and Huang, 2013*). Consistent with the increased TAG in tissues, Nile red staining of the FBs of *trpγ¹* and *trpγ^G4* flies exhibited larger lipid mass compared to control animals under sated condition (*Figure 2F–I*). These LDs returned to their normal size through the expression of *trpγ* in *trpγ* or *Dh44* neurons but not *dILP2* neurons (*Figure 2F and J–L*). Additionally, inactivating the *Dh44* cells (*Dh44-GAL4/UAS-Kir2.1*) recapitulated the enlarged LD phenotype of the *trpγ* mutant flies (*Figure 2—figure supplement 1D–G*). Overall, our findings indicated that *trpγ* expression regulates lipid and carbohydrate homeostasis but not protein levels in DH44 neuroendocrine cells in the PI.

## Trpγ mutants exhibit starvation susceptibility and deficits in lipolysis

Higher lipid cellular levels may decrease lifespan (*Johnson and Stolzing, 2019*; *Tatar et al., 2014*). Therefore, we measured the lifespan of control and *trpγ¹* flies fed with a standard cornmeal diet (*Figure 3A*). However, the lifespans of control and *trpγ¹* flies were not significantly different under normal conditions. The LT$_{50}$ of the control was 59.34±0.92 d and that of *trpγ¹* was 55.75±2.35 d. In contrast, *trpγ* appeared to be required for proper metabolism under starvation conditions, as demonstrated by the decreased starvation resistance of the *trpγ¹* and *trpγ^RNAi* knockdown flies in *Dh44* cells (*Figure 3B*; *Dhakal et al., 2022*). This defect was fully recovered by the expression of *UAS-trpγ* under the control of *Dh44-GAL4* (*Figure 3B*).

The starvation-sensitive phenotype in the *trpγ* mutants may have been due to decreased carbohydrate storage in tissues, including glucose, trehalose, and glycogen. However, we previously demonstrated that *trpγ* mutants could utilize carbohydrates under starved conditions (*Dhakal et al., 2022*). Furthermore, the elevated whole-body TAG levels in the *trpγ* mutant flies might suggest that they were unable to break down stored lipids even under starving conditions. Metazoans must coordinate the metabolism of glycogen, lipid, and protein to maintain metabolic homeostasis during fasting periods, thus providing an appropriate energy supply across tissues. Therefore, the total TAG levels of the control and *trpγ* mutant flies were assessed under sated and starved (starvation for 24 hr) conditions (*Figure 3C–F*). When the control flies were deprived of food for 24 hr, their TAG levels decreased significantly (*Figure 3C*). In contrast, the *trpγ¹* mutant exhibited no changes in whole-body TAG levels before and after starvation (*Figure 3C*). To further confirm that lipolysis was restricted in the *trpγ* mutants under starvation conditions, LDs were stained, after which we measured the sizes of individual LDs accumulated throughout the whole fat bodies of the flies (*Figure 3D–F*). Despite considerable variations, the sizes of the LDs in the control flies were significantly reduced after starvation. In

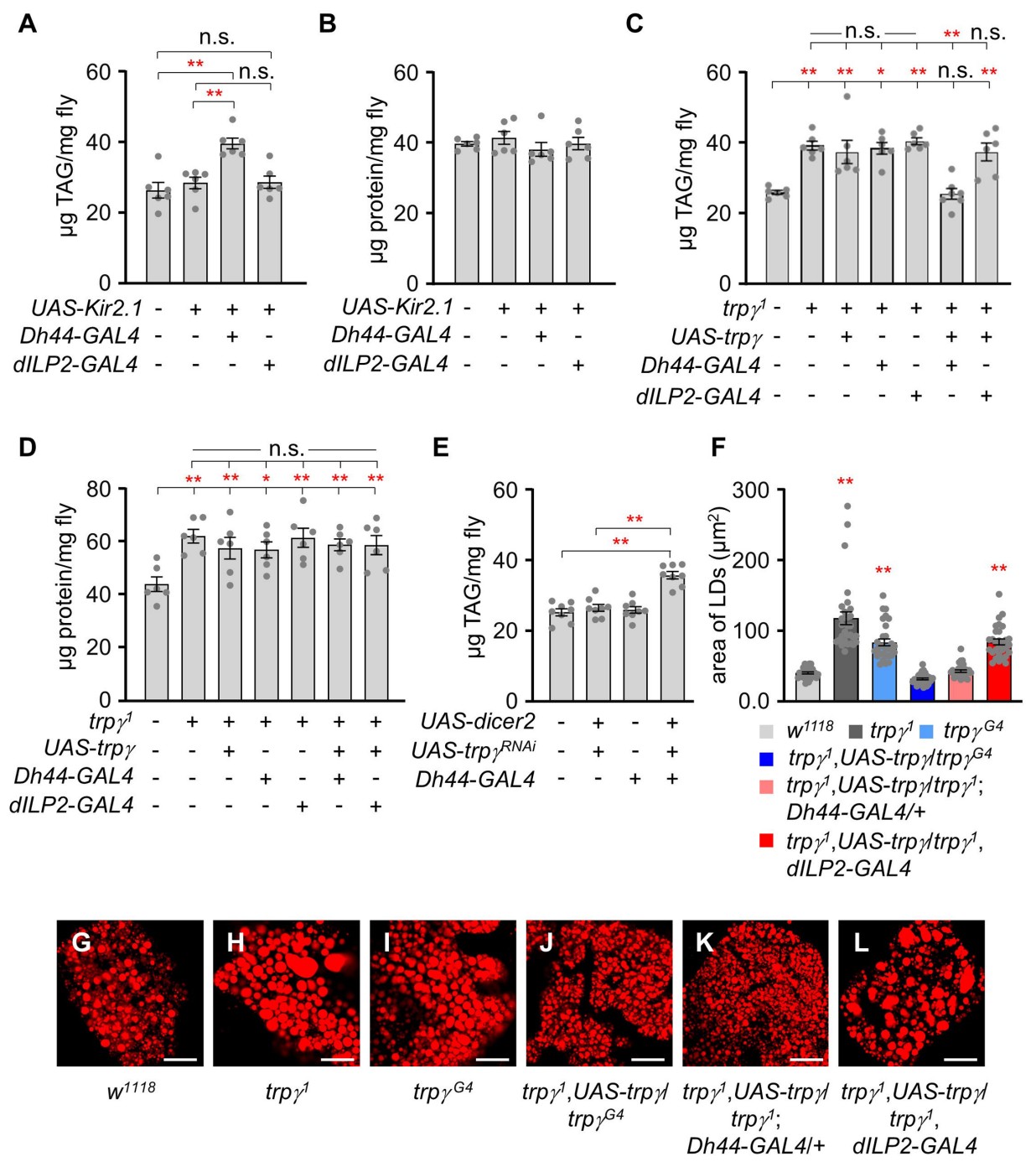

**Figure 2.** *Dh44* neurons are essential for regulating lipid tissue levels. (**A**) Tissue triacylglycerol (TAG) level measurement in whole-body extracts of adult male flies after silencing of *Dh44-GAL4* and *dILP2-GAL4* with *UAS-Kir2.1* (n=6). (**B**) Measurement of tissue protein level in whole-body extracts of adult male flies. Brain-specific *Dh44-GAL4* and *dILP2-GAL4* neurons were ablated using *UAS-Kir2.1* (n=6). (**C**) Measurement of TAG from whole-body extract of adult male flies in the indicated genotypes (n=6–7). (**D**) Measurement of tissue protein level from the whole-body extracts of adult males (n=6). (**E**) TAG levels in whole-body extracts after RNAi knockdown of *trpγ* mutants under control of the *Dh44-GAL4* (n=6). (**F**) Measurement of area (µm²) of LDs in adult fat body across the indicated genotypes involved the selection of the 30 largest lipid droplets (LDs), choosing the top 10 LDs from each sample for analysis (n=3). (**G-L**) Nile red stating of fat body from the male of indicated genotypes. Scale bars represent 50 µm. All values are reported as means ± SEM. Comparisons between multiple experimental groups were conducted via single-factor ANOVA coupled with Scheffe's *post hoc* test. The asterisks indicate significant differences from the controls (*p<0.05, **p<0.01). Each dot indicates the distribution of individual sample values. (+) and (-) indicate the presence or absence of the indicated transgenes, respectively.

The online version of this article includes the following figure supplement(s) for figure 2:

**Figure supplement 1.** Expression of trpγ, Dh44 and dilp2 in the brain of *Drosophila*.

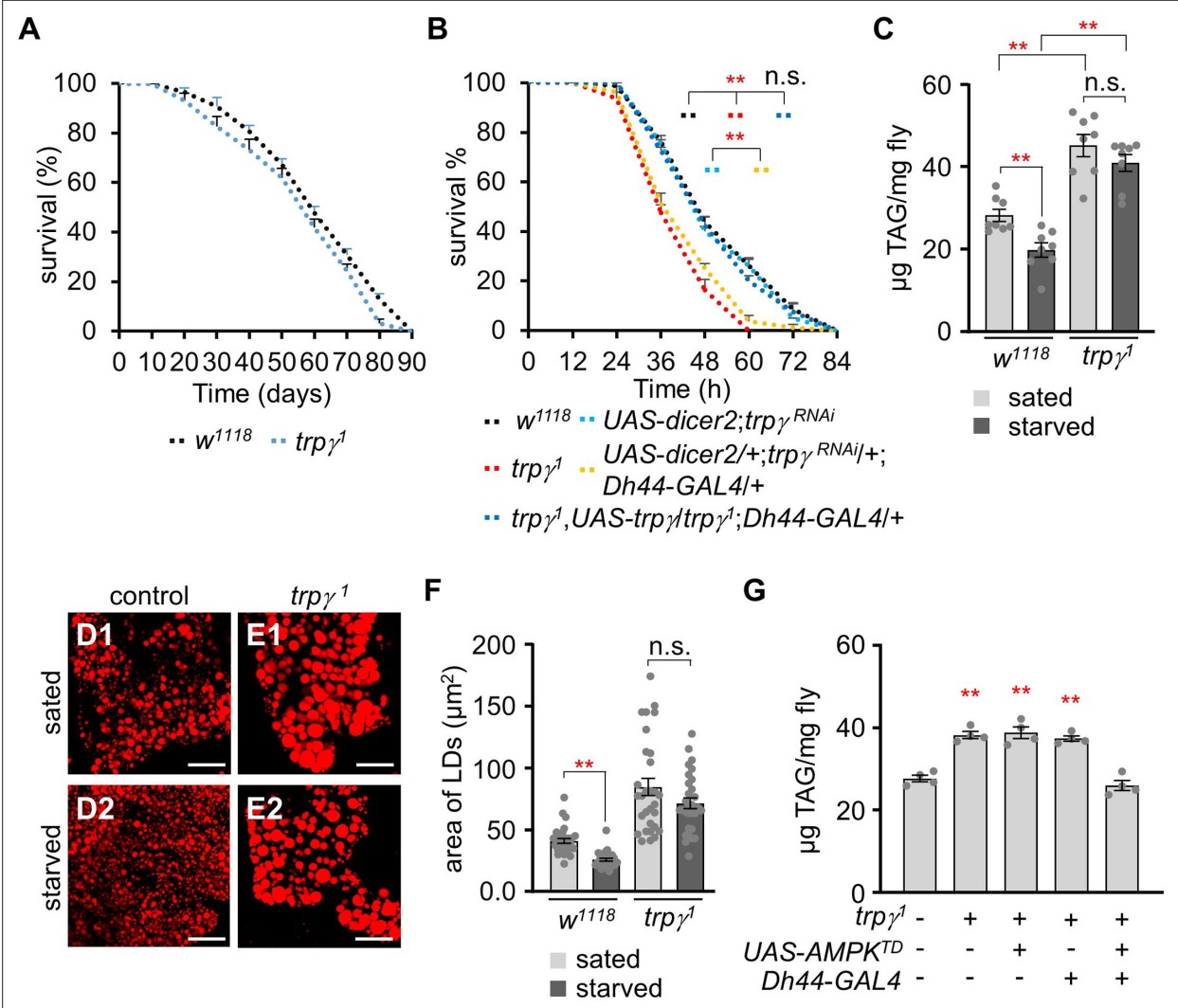

**Figure 3.** *trpγ¹* have deficits in the lipolytic pathway under starvation conditions. (**A**) Survival assay to measure the total survival time (days) of control (*w¹¹¹⁸*) and *trpγ¹* male flies fed with a normal corn meal diet (n=7–8). (**B**) Survival assay to measure the survival time (h) of the indicated genotypes with male flies under starvation conditions (n=4). (**C**) TAG level measurement in control (*w¹¹¹⁸*) and *trpγ¹* adult male flies in both sated (0 hr starvation) and starved (24 hr starvation) conditions (n=8). (**D, E**) Nile red staining of the lipid droplets (LDs) extracted from FB of (**D**) *w¹¹¹⁸* and (**E**) *trpγ¹* flies under sated (**D1, E1**) and starved (**D2, E2**) conditions, respectively. Scale bars represent 50 μm. (**F**) Measurement of area (μm²) of the LDs extracted from the FBs of *w¹¹¹⁸* and *trpγ¹* flies under sated (0 hr starvation) and starved (24 hr starvation) conditions (n=3). (**G**) Measurement of triacylglycerol (TAG) level with controls and the flies after expressing *UAS-AMPKᵀᴰ* under the control of *Dh44-GAL4* in the *trpγ¹* mutant background (n=4). All values are reported as means ± SEM. Survival curves in A and B were estimated for each group, using a Kaplan-Meier method and compared statistically using the log-rank tests. Comparisons between multiple experimental groups in C, F, G, and H were conducted via single-factor ANOVA coupled with Scheffe's *post hoc* test. The asterisks indicate significant differences from the controls (**p<0.01).

The online version of this article includes the following figure supplement(s) for figure 3:

**Figure supplement 1.** Measurement of triacylglycerol (TAG) levels after overexpressing *UAS-trpγ* and knock-down of *trpγᴿᴺᴬⁱ* in *Dh44* neurons.

contrast, no significant differences in LD sizes were identified between the sated and starved conditions in the *trpγ¹* flies. We examined how overexpression and knockdown of *trpγ* in *Dh44* neurons affect the starvation phenotype. Overexpressing *trpγ* in *Dh44* cells resulted in similarity to the wild-type in both sated and starved conditions, as well as normal survival time under starvation conditions (*Figure 3B* and *Figure 3Figure 1A*). Conversely, *trpγᴿᴺᴬⁱ* knockdown flies in *Dh44* neurons reproduced phenotypic traits observed in *trpγ* mutants, including decreased lipid levels (*Figure 3—figure supplement 1B*) and reduced survival time under starvation conditions (*Figure 3B*).

*Dh44* neurons regulate starvation-induced sleep suppression (*Oh and Suh, 2023*), which implies that these neurons become more active under starved conditions. Adenosine monophosphate-activated protein kinase (AMPK) serves as the master controller for maintaining energy balance in cells, and coordinating metabolic pathways (*Hardie et al., 2003*). It regulates the balance between building up and breaking down substances to ensure cellular stability during metabolic stress. AMPK is a key target for treating metabolic diseases like type 2 diabetes and obesity, as its activation increases fatty acid oxidation (*Kim et al., 2016*). We wonder if activating AMPK in *Dh44* neurons improves lipolysis. The expression of AMPK[TD], the activated form of AMPK (*Lee et al., 2007*) in *Dh44* neurons indeed restored the elevated TAG levels observed in *trpγ¹* (*Figure 3G*). This suggests that AMPK functions as a downstream component of *Dh44* neuronal activation.

## Recovery of the starvation resistance of the trpγ mutant via metformin treatment

Metformin is widely used to treat many metabolic diseases such as type II diabetes (*Lv and Guo, 2020*). We previously demonstrated that 1–5 mM metformin can induce hypoglycemia, in addition to suppressing fat storage in flies (*Nath and Lee, 2025*; *Sang et al., 2021*). Therefore, we next sought to test whether the increased lipid levels of the *trpγ* mutant can be reduced to a normal level via oral administration of metformin. First, we measured TAG levels at 0, 7, and 14 d after dietary administration of 1 mM or 5 mM of metformin (the treatments were prepared by mixing the appropriate metformin concentrations into standard cornmeal diets) (*Figure 4A and B*). The high TAG levels in the *trpγ* mutant (41.91± 4.56) were decreased to levels similar to those of the control at 7 and 14 d (30.03± 4.34 and 33.79± 3.20, respectively) after treatment with 1 mM metformin (*Figure 4A*). Furthermore, although oral administration of 1 mM metformin did not affect the TAG level in the control, the 5 mM metformin treatment significantly reduced the TAG level of the control flies at 14 d (*Figure 4B*). Moreover, the 5 mM metformin treatment was more effective in reducing the TAG level in the *trpγ* mutant. These findings indicated that oral administration of metformin can either suppress lipogenesis or enhance lipolysis in *D. melanogaster*. To further confirm the effects of metformin treatment, LDs were analyzed under the same condition (*Figure 4C and D*). The sizes of the LDs in the control and *trpγ* mutant flies were significantly reduced after treatment with 1 mM metformin at 14 d or 5 mM metformin at 7 and 14 d (*Figure 4D*).

Next, we sought to assess whether metformin could increase starvation resistance in the control and *trpγ* mutant flies. To test this hypothesis, the starvation resistance of the flies was measured with or without metformin. Neither of the metformin concentrations appeared to affect the starvation resistance of the control flies (*Figure 4E and F*). However, although metformin did not restore the starvation resistance of the *trpγ* mutant flies to the level of the controls, it did significantly extend their survival (*Figure 4E and F*; $LT_{50}$ of the control at 1 mM and 5 mM: 56.06±1.40 hr and 52.39±2.09 hr, respectively; $LT_{50}$ of the *trpγ¹* mutant at 1 mM and 5 mM: 50.35±3.34 hr and 46.42±1.54 hr, respectively). Our findings thus demonstrated that oral metformin administration can rescue TRPγ-mediated metabolic syndrome.

## Downregulation of the lipolytic gene *brummer* in *trpγ* mutants

The *trpγ¹* mutant exhibited marked alterations in the levels of major nutrients. Therefore, we next sought to analyze the transcriptional levels of genes related to gluconeogenesis, lipogenesis, and lipolysis. Gluconeogenesis is a metabolic process that produces glucose from non-carbohydrate carbon substrates. The genome of *D. melanogaster* harbors two gluconeogenic genes: *fructose-1,6-bisphosphatase* (*fbp*) and *phosphoenolpyruvate carboxykinase 1* (*pepck1*) (*Miyamoto and Amrein, 2019*). The results of our real-time quantitative reverse transcription PCR (qRT-PCR) analyses indicated that the transcription levels of *fbp* and *pepck1* were not significantly different between control flies under sated and starved conditions (*Figure 5A*). Next, we analyzed two lipogenic genes: *acetyl–CoA carboxylase* (*acc*) and *desaturase 1* (*desat1*) (*Figure 5A*; *Wang et al., 2022*; *Yee et al., 2013*). No significant differences were identified between the control and *trpγ¹* flies under both conditions. Finally, we investigated the expression of the lipolytic gene *brummer* (*bmm*) (*Figure 5A*; *Blumrich et al., 2021*). Under starved conditions, the transcriptional level of *bmm* was highly increased in the control. In contrast, *bmm* was downregulated in *trpγ¹* under sated and starved conditions. Therefore, the transcriptional regulation of *w¹¹¹⁸* and *trpγ¹* during starvation was markedly and significantly different.

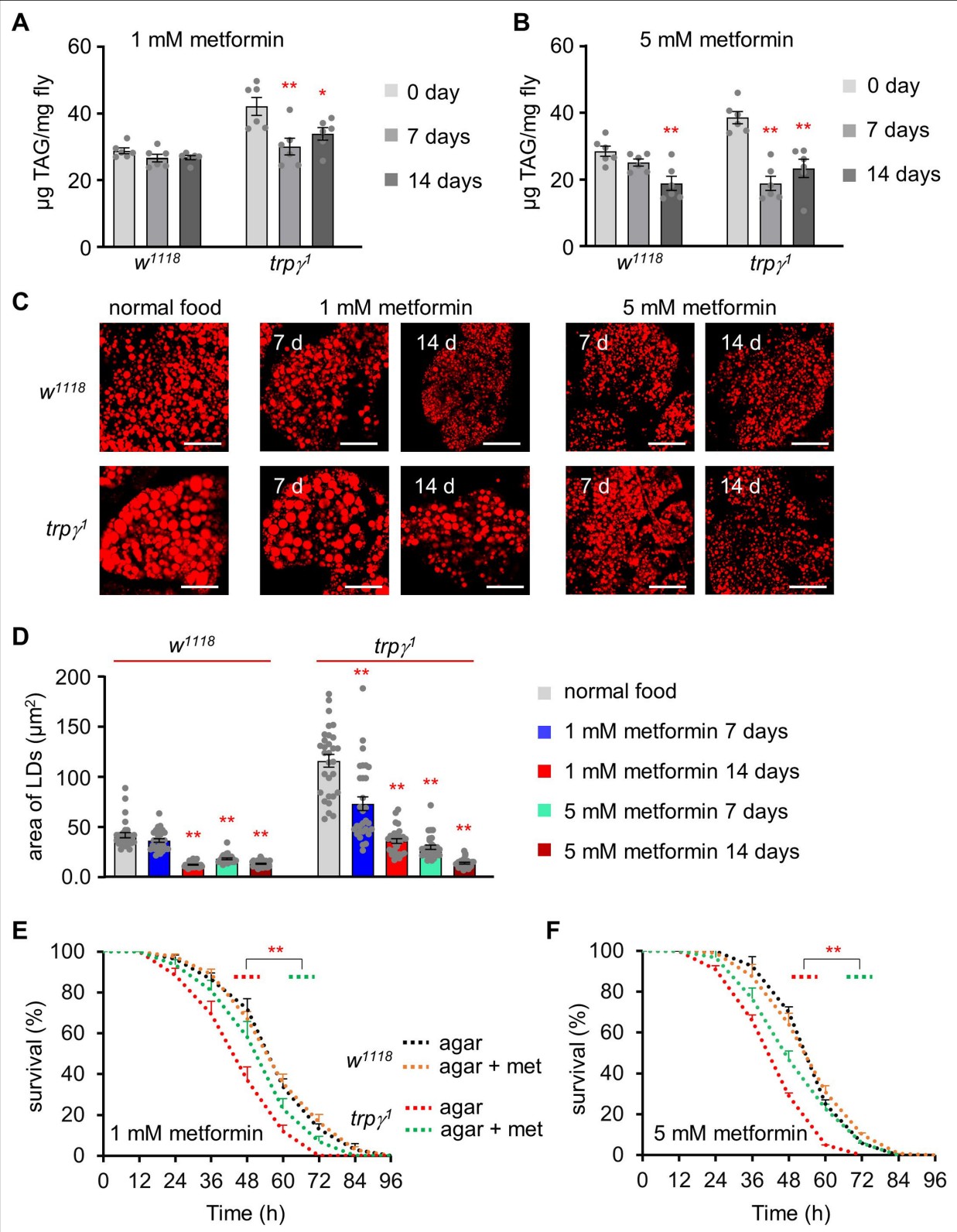

**Figure 4.** Rescue of starvation susceptibility phenotype using the lipolytic drug metformin. (**A**) Triacylglycerol (TAG) level measurement at 0, 7, and 14 d in the control (*w^1118*) and *trpγ^1* adult male flies after dietary exposure to 1 mM metformin (n=6). (**B**) TAG measurement at 0, 7, and 14 d in *w^1118* and *trpγ^1* adult male flies after dietary exposure to 5 mM metformin (n=6). (**C**) Pictures of Nile red staining of lipid droplets (LDs) after dietary exposure to 1 mM and 5 mM metformin in standard fly food for 7 and 14 d in *w^1118* and *trpγ^1* flies. Scale bars represent 50 μm. (**D**) Measurement of area (μm²) of LDs

*Figure 4 continued on next page*

*Figure 4 continued*

extracted from the FB of *w^1118* and *trpγ^1* flies after dietary exposure to 1 mM and 5 mM metformin in standard fly food for 7 and 14 d (n=3). (**E**) Survival assay to measure the survival time (h) of control (*w^1118*) and *trpγ^1* males after dietary exposure to 1 mM metformin in 1% agar food (n=6–10). (**F**) Starvation survival assay of control (*w^1118*) and *trpγ^1* males after dietary exposure to 5 mM metformin in 1% agar food (n=6). All values are reported as means ± SEM. Comparisons between multiple experimental groups were conducted via single-factor ANOVA coupled with Scheffe's *post hoc* test. Each dot indicates the distribution of individual sample values. Survival curves in E and F were estimated for each group, using a Kaplan-Meier method and compared statistically using the log-rank tests. The asterisks indicate significant differences from the controls (*p<0.05, **p<0.01).

The data indicates that the *trpγ* mutant can sense the starvation state but responds abnormally by suppressing lipolysis instead of activating it. This dysregulated lipolytic response likely increases the mutant's vulnerability to starvation, as it cannot effectively mobilize lipid stores for energy during periods of nutrient deprivation. Brummer, a homolog of human adipocyte triglyceride lipase (ATGL), is associated with lipid storage in the form of LDs (*Grönke et al., 2005*; *Men et al., 2016*). These results suggest that the lipolytic process is not sufficient to deplete stored lipids in the *trpγ^1* mutant, resulting in excessive lipid storage. Therefore, the survival time of the *trpγ^1* mutant during starvation was significantly shorter than that of the control.

Transgenic flies expressing *bmm::GFP* (*bmm* promoter *GFP*) in their FB cells were used to examine *bmm* expression in LDs (*Men et al., 2016*). The fluorescent images obtained from the transgenic flies were then used to confirm our qRT-PCR results. Two images of control flies under sated and starved conditions were first analyzed both qualitatively and quantitatively (*Figure 5B and C*). Next, *trpγ^1* was analyzed under the two conditions (*Figure 5B and C*). A reduction in fluorescent intensity was clearly detectable in the *trpγ^1* mutant using the transgenic reporter line. These results further confirmed that *trpγ^1* has aberrant regulation in the lipolytic pathway.

To further substantiate the idea that the reduction in the expression level of *bmm* is the primary cause of the enlarged LDs in *trpγ* mutant flies, we measured LDs in the FB after expressing *UAS-bmm* using specific *GAL4* drivers: *trpγ^G4*, *Dh44-GLA4*, *r4-GAL4* (fat body specific), *and Myo1A-GAL4* (gut enterocyte specific). Only *trpγ^G4*, *Dh44-GAL4,* and *r4-GAL4* were able to restore the enlarged LDs defects in the FB, as well as TAG levels of the whole body, after expressing *UAS-bmm* (*Figure 5D–J* and *Figure 5—figure supplement 1*). These findings suggest that the downregulation of *bmm* is one of the factors contributing to the elevated lipid defects in *trpγ^1*. Furthermore, the expression of *bmm* in the fat body, as well as *Dh44* neurons in the PI region, can promote lipolysis at the systemic level.

To conclusively explore whether metformin targets enzymes involved in lipid metabolism, specifically lipolytic or lipogenic genes, we administered 5 mM metformin supplemented in normal food for 1 d and measured the expression levels of *bmm*, *acc*, and *desat1*. Oral supplementation of metformin resulted in an increase in the expression level of the lipolytic gene, *bmm* lipase of the control as well as *trpγ^1* (*Figure 5K*). Importantly, there was no observed effect of metformin on the expression levels of lipogenic genes. This data provides insights into the mode of action of metformin, suggesting a specific impact on lipolysis.

## Recovery of the trpγ mutant through oral lipase administration or lipid absorption

The intestine is a key organ for lipid absorption and metabolism (*Han et al., 2020*) and disturbances in intestinal lipid metabolism are associated with hyperlipidemia (*Jia et al., 2021*). To clarify if our observations were due to insufficient activity of lipase in the gut, lipase enzyme mixed with 1% agar food was supplied to both control and mutant flies. Given that the brief activity of lipase in the intestine, lipase-containing agar food was provided to the flies every 12 hr. Surprisingly, supplementation with lipase, but not denatured lipase, under starvation conditions increased the survival of *trpγ* mutant flies to levels comparable to those of the control flies (*Figure 6A*; LT$_{50}$ of lipase treated-control and *trpγ^1*: 50.35±0.93 hr and 44.94±1.33 hr, respectively). To support the hypothesis that the extended lifespan of mutant flies during starvation is associated with dietary lipase intake, we examined lipid accumulation (*Figure 6—figure supplement 1A–E*). Lipase treatment resulted in the breakdown of lipid deposits in the mutant R2 region of the intestine. Importantly, only active lipase showed an enhanced survival time under starvation conditions in mutant flies, while denatured lipase did not exhibit this effect. Therefore, our findings suggest that lipase may play a role in digesting accumulated lipids in the intestine, though the mechanisms of lipase absorption remain unclear.

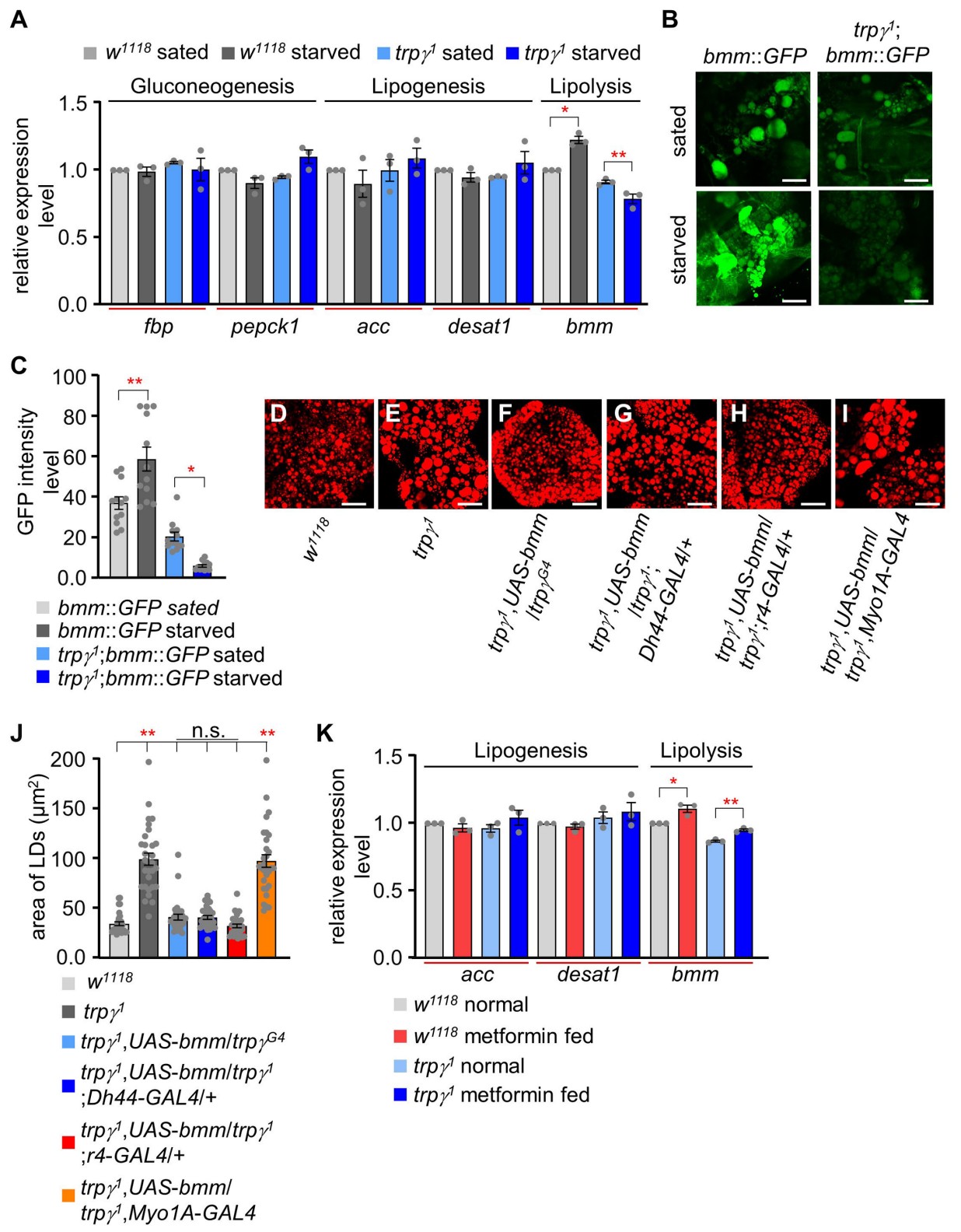

**Figure 5.** Quantitative analysis of the expression of gluconeogenic, lipogenic, and lipolytic genes and the effect of *bmm* expression and metformin feeding. (**A**) qRT-PCR analysis to measure the expression of gluconeogenic genes (*fbp*, *pepck1*), lipogenic genes (*acc*, *desat1*), and a lipolytic gene (*bmm*) under sated (0 hr starvation) and starved (24 hr starvation) conditions in *w[1118]* and *trpγ[1]* flies. The relative fold change in the expression of starvation-induced genes (gluconeogenic genes: *fbp* and *pepck1*; lipogenic genes: *acc* and *desat1*; and lipolytic gene: *bmm*) was determined in whole-

*Figure 5 continued on next page*

*Figure 5 continued*

body samples of male adult *w^1118* and *trpγ^1* flies by qRT-PCR. *Tubulin* was used as an internal control to standardize the samples. Each graph shows the number of evaluated samples (n=3). (**B**) Fluorescence microscopic imaging of *bmm::GFP* expression in the FBs of *w^1118* and *trpγ^1* under sated (0 hr starvation) and starvation (24 hr starvation) conditions. Scale bars represent 50 µm. (**C**) Quantification of intensity level of *bmm::GFP* in the FBs of *w^1118* and *trpγ^1* under sated and starvation conditions (n=3). (**D-I**) Nile red staining of lipids in the FB of flies with the indicated genotypes. Scale bars represent 50 µm. (**J**) Measurement of area of lipid droplets (LDs) from samples D–I entailed selecting a total of 30 LDs, with the 10 largest LDs chosen from each sample for analysis (n=3). (**K**) qRT-PCR analysis to measure the expression of *acc*, *desat1*, and *bmm* from the whole body samples of flies after feeding 5 mM metformin for 1 d (n=3). All values are reported as means ± SEM. Comparisons between multiple experimental groups were conducted via single-factor ANOVA coupled with Scheffe's *post hoc* test. The asterisks indicate significant differences from the controls (*p<0.05, **p<0.01).

The online version of this article includes the following figure supplement(s) for figure 5:

**Figure supplement 1.** Measurement of triacylglycerol (TAG) levels after expressing *UAS-bmm* in *trpγ^G4*, *Dh44-GAl4*, *r4-GAL4*, and *Myo1A-GAL4* (n=4).

Next, we sought to assess the involvement of each of the components of TAG in the recovery of starvation susceptibility. TAG is composed of glycerol and three fatty acids. Therefore, we first fed the flies with 1% glycerol only. Glycerol extended the survival of the control and *trpγ^1* flies by 3–4-fold (**Figure 6B**). However, the $LT_{50}$ of the control and *trpγ^1* were still significantly different (**Figure 6C**; $LT_{50}$ of control and *trpγ^1*: 244.5± 20.10 hr and 117.75± 3.32 hr, respectively). Next, the flies were fed with 0.2% and 0.5% hexanoic acid (HA) (**Figure 6C** and **Figure 6—figure supplement 1F**; 0.2% HA, $LT_{50}$ of control and *trpγ^1*: 65.44±1.94 hr and 52.32±2.05 hr, respectively; 0.5% HA, $LT_{50}$ of control and *trpγ^1*: 56.27±1.30 hr and 46.27±1.47 hr, respectively). Additionally, we also tested the effects of 0.2% and 0.5% concentrations of a TAG mixture (combination of mono-, di-, and tri- acylglycerol) (**Figure 6D** and **Figure 6—figure supplement 1G**; 0.2% TAG, $LT_{50}$ of control and *trpγ^1*: 52.93±5.12 hr and 50.1±3.74 hr, respectively; 0.5% TAG, $LT_{50}$ of control and *trpγ^1*: 65.1±2.6 hr and 57.75±3.32 hr, respectively). The HA and mixed TAG treatments significantly extended the survival of the *trpγ^1* mutants, albeit not to levels similar to those of the control. This indicated that the TRPγ mutant could not fully absorb and burn the digested lipid. Furthermore, lipid-only dietary supplementation markedly limited the $LT_{50}$ of the control. Importantly, supplementation with lipolytic drugs, lipase, TAG, and free fatty acids effectively rescued the survival of *trpγ* mutants under starvation conditions. This observation was consistent with our finding that *trpγ* mutants are unable to degrade lipid stores under starvation conditions. The dietary rescue of *trpγ* mutants suggests that although the *trpγ* mutation increases the levels of stored lipids, the flies were not able to utilize them when starved. We next sought to determine whether *trpγ* mutation reduces lipase production in the abdomen by comparing lipid accumulation in the intestine. The *Drosophila* midgut is divided into R1–R5 regions, each with distinct functions in nutrient processing and absorption. Notably, the R2 region serves as a critical interface between the fly's diet and metabolic processes (**Capo et al., 2019**). To assess lipid distribution, we conducted Nile red staining in control and mutant animals (**Figure 6E–H**). Mutant flies displayed elevated lipid deposits on the intestinal wall of the R2 region (**Figure 6F**), a phenotype restored to normal levels by expressing *trpγ* in its own cells or *Dh44* neurons (**Figure 6G–I**). Consequently, we quantified lipid droplets specifically within the intestinal wall of the R2 region in *trpγ^1* flies, with no such accumulation observed in control or rescued flies (**Figure 6I**). These findings indicate that TRPγ plays a crucial role in the brain-gut axis, specifically in controlling lipid metabolism in the intestine, as evidenced by the lipid accumulation observed in the gut region of *trpγ^1*. Taken together, these observations indicate that TRPγ plays an important role in maintaining systemic lipid levels through proper expression of triglyceride lipase.

## Assessment of the role of DH44 and its receptors (DH44R1 and DH44R2) in relation to the utilization of fat

In *Drosophila*, two receptors have been identified for DH44 (**Hector et al., 2009**). In order to investigate the potential roles of DH44 in fat utilization and identify the receptor responsible for nutrient regulation, we examined lipid and protein levels in *Dh44^Mi*, *Dh44R1^Mi*, and *Dh44R2^Mi*. *Dh44^Mi* and *Dh44R1^Mi* showed normal total TAG levels, while *Dh44R2^Mi* exhibited higher lipid levels like *trpγ^1* (**Figure 7A**). Additionally, Nile red staining of LDs in the FBs revealed that *Dh44^Mi* and *Dh44R2^Mi* mutants possessed larger LDs (**Figure 7B–F**), although protein levels remained unaffected across all three mutants (**Figure 7G**). Further investigations quantified the expression of the *bmm* in the FBs of *Dh44^Mi*, *Dh44R1^Mi*, and *Dh44R2^Mi* under both sated and starved conditions. In sated conditions,

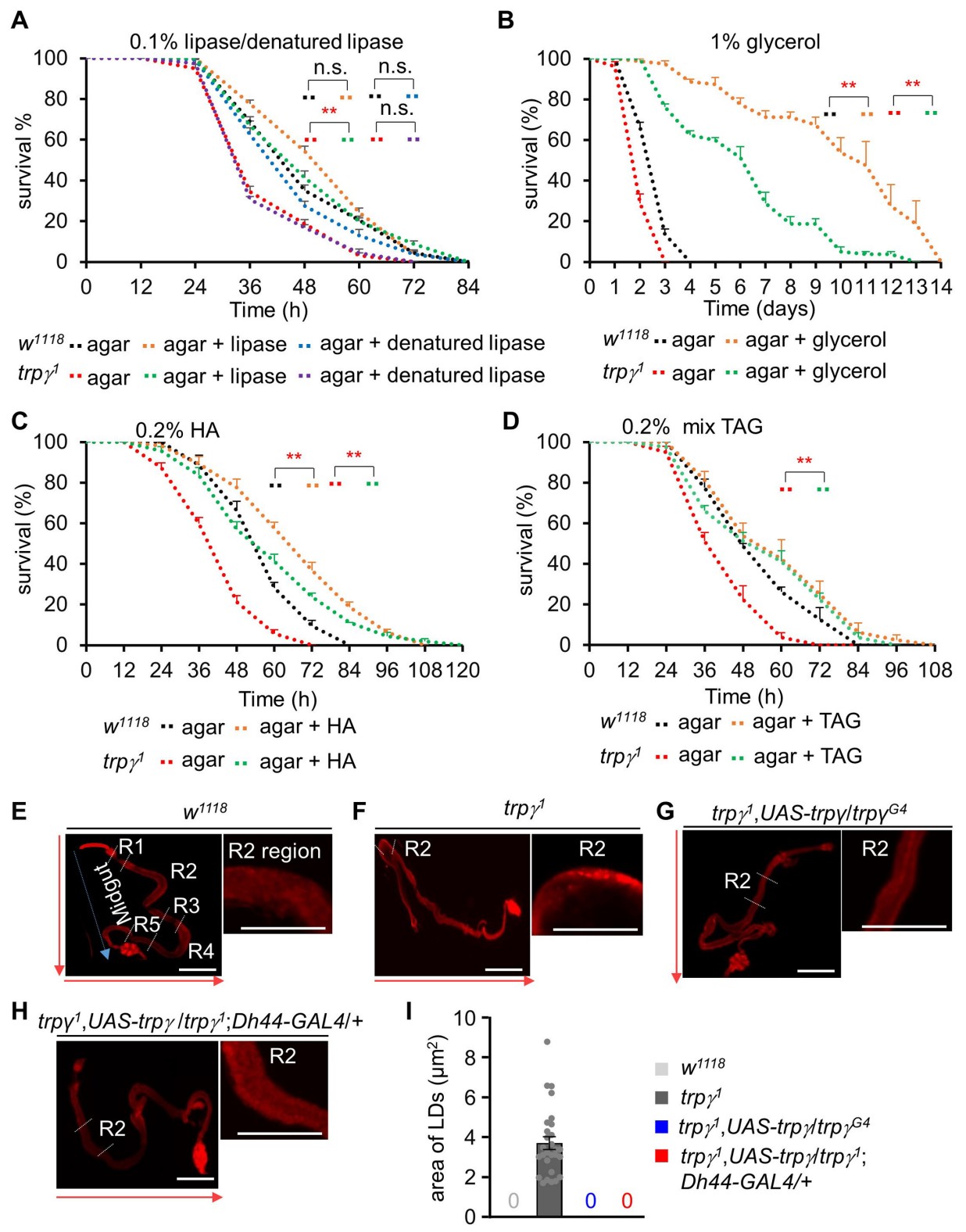

**Figure 6.** Dietary supply of lipase, glycerol, mixed triacylglycerol (TAG), and free FA to rescue the starvation sensitivity phenotype. (**A**) Survival assays of $w^{1118}$ and $trp\gamma^1$ flies under starvation condition by feeding 0.1% lipase or 0.1% denatured lipase mixed into 1% agar food (n=8). (**B**) Starvation survival assay to measure the survival time (h) of $w^{1118}$ and $trp\gamma^1$ male flies after feeding 1% glycerol mixed into 1% agar food (n=4). (**C**) Survival assay to measure the survival time (h) of $w^{1118}$ and $trp\gamma^1$ flies after feeding 0.2% hexanoic acid (HA) supplemented into 1% agar food (n=8). (**D**) Survival assay to measure the survival time (hr) of $w^{1118}$ and $trp\gamma^1$ flies after feeding 0.2% mixed (mono-, di-, and tri-) glycerides (n=4). (**E-I**) Nile red staining of LDs in full gut and the

*Figure 6 continued on next page*

*Figure 6 continued*

magnified R2 region of adult male flies. (**E**) *w^1118^*, (**F**) *trpγ^1^*, (**G**) *trpγ^1^,UAS-trpγ/trpγ^G4^*, (**H**) *trpγ^1^,UAS-trpγ/trpγ^1^;Dh44-GAL4/+*. Scale bars represent 50 μm. The arrow indicates the orientation of intestine from anterior to posterior. (**I**) Measurement of area of lipid droplets (LDs) in the R2 region of *trpγ^1^*, area of 28 LDs from 10 samples (n=10). Note that control and rescued flies have no LD. All values are reported as means ± SEM. Survival curves were estimated for each group, using a Kaplan-Meier method and compared statistically using the log-rank tests. (**p<0.01).

The online version of this article includes the following figure supplement(s) for figure 6:

**Figure supplement 1.** Measurement of lipase effect in lipid droplets (LDs) and starvation survival assay after feeding hexanoic acid or mixed triacylglycerol (TAG).

*bmm* expression was found to be downregulated in *Dh44R2^Mi^* compared to the control (*Figure 7H*). Interestingly, under starved conditions, while *bmm* levels significantly increased in the control, indicating a normal response to starvation, such an increase was not observed in *Dh44^Mi^* and *Dh44R2^Mi^*. Instead, in these mutants, *bmm* levels significantly decreased compared to their levels under sated conditions (*Figure 7H*). In addition, *bmm* levels in *Dh44R1^Mi^* under starved conditions did not increase as significantly as in the control. This suggests a unique role of DH44 and its receptors in regulating lipid metabolism and response to nutritional status in *Drosophila*.

To further verify whether *Dh44*, *Dh44R1*, and *Dh44R2* mutants can be recovered by oral lipase administration or lipid absorption, we measured the starvation survival time with 1% agar and 1% agar with 0.1% lipase, 1% glycerol, 0.2 % HA, and 0.2% mix TAG (*Figure 7—figure supplement 1A–D*). Under starvation conditions, *Dh44^Mi^* and *Dh44R2^Mi^* showed significantly reduced survival time compared to control, whereas *Dh44R1^Mi^* flies were normal (LT$_{50}$ of control, *Dh44^Mi^*, *Dh44R1^Mi^*, and *Dh44R2^Mi^*: 54.32±2.29 hr, 36.95±1.76 hr, 46.83±1.27 hr, and 36.33±1.03 hr, respectively). Interestingly, the reduced survival time observed in *Dh44^Mi^* and *Dh44R2^Mi^* mutants were improved with the supplementation of lipase (LT$_{50}$ of *Dh44^Mi^* and *Dh44R2^Mi^*: 54.72±0.55 h and 48.63±1.18 hr), glycerol (LT$_{50}$ of *Dh44^Mi^* and *Dh44R2^Mi^*: 239±1.91 hr and 238.5±5.12 hr), hexanoic acid (LT$_{50}$ of *Dh44^Mi^* and *Dh44R2^Mi^*: 62.50±1.50 hr and 58.77±0.81 hr), and mixed TAG (LT$_{50}$ of *Dh44^Mi^* and *Dh44R2^Mi^*: 54.70±3.53 hr and 59.50±2.06 hr) (*Figure 7—figure supplement 1A–D*).

To explore the connection between TRPγ and DH44 signaling, we expressed *UAS-trpγ* under the control of *Dh44R1-GAL4* and *Dh44R2-GAL4*. Only *Dh44R2-GAL4* restored the lipid defect in *trpγ^1^*, unlike *Dh44R1-GAL4* (*Figure 7I*). Furthermore, *Dh44R2-GAL4*, but not *Dh44R1-GAL4*, rescued the excessive lipid accumulation in the FB of *trpγ^1^* mutant (*Figure 7J–N*). Given that *Dh44R2* is predominantly expressed in the intestine, we performed immunostaining to examine whether *Dh44R2* co-localizes with *trpγ* in gut cells. Our results confirmed that *Dh44R2* and *trpγ* are co-expressed in intestinal cells (*Figure 7O and P*). Additionally, we analyzed *Dh44R2* expression in the brain and found that two *Dh44R2*-expressing cells are co-localized with *Dh44*-expressing cells in the PI region (*Figure 7Q*). To further delineate whether *Dh44R2*-mediated fat utilization is specific to the brain, gut, or fat body, we knocked down *Dh44R2^RNAi^* using *Dh44-GAL4*, *Myo1A-GAL4*, and *cg-GAL4* (*Lee et al., 2018a*), respectively (*Figure 7—figure supplement 1E*). Notably, the knockdown of *Dh44R2* with *Myo1A-GAL4* resulted in elevated TAG levels, indicating that DH44R2 activity in lipid metabolism is specific to the gut.

## Discussion

Here, we found that the function of TRPγ, one of the TRPC channels of *D. melanogaster*, in DH44 neuroendocrine cells plays an essential role in lipid regulation, which is linked to alterations in membrane lipids (*Cobb et al., 2021*; *Overgaard et al., 2005*). The *trpγ^1^* mutant exhibited clear signs of metabolic syndrome, as demonstrated by reduced carbohydrate levels coupled with much higher protein and lipid levels in the body (*Figure 8*). More importantly, we found that the expression of *trpγ* in DH44 neurons was necessary and sufficient to regulate the carbohydrate and lipids. However, we failed to identify the specific cells required for regulating protein homeostasis because the increased protein levels in the *trpγ* mutants were not recovered by pilot screening (data not shown) except in the *trpγ-GAL4*. In conclusion, our findings suggested that the DH44 system has an important role in regulating the metabolic homeostasis of carbohydrates and lipids (*Figure 8*).

The metabolic dysregulation of carbohydrates and lipids observed in the *trpγ* mutants was phenocopied by the cellular inactivation of DH44 cells. Compared to the controls, the *Dh44-GAL4/UAS-Kir2.1*

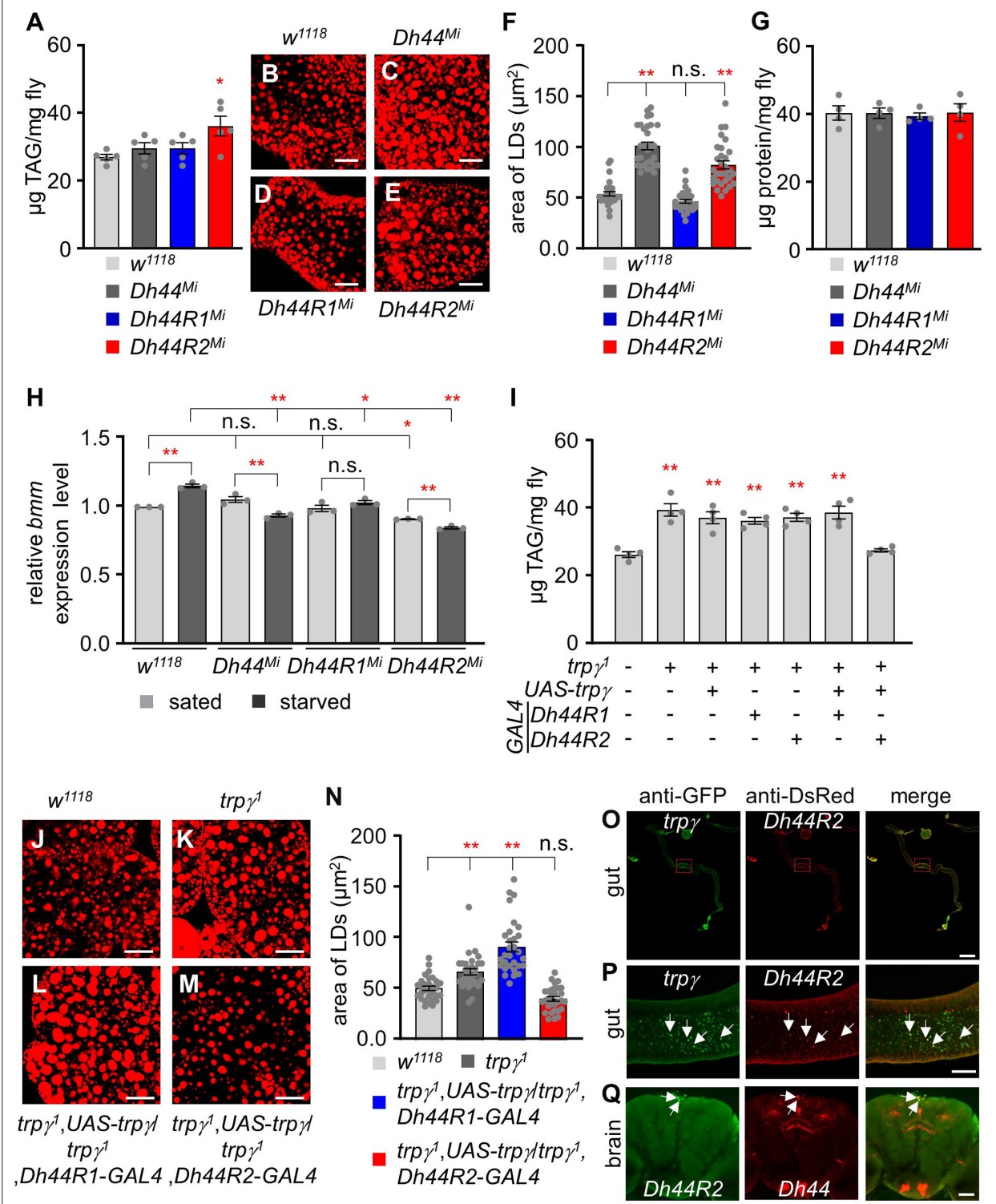

**Figure 7.** Functional analysis of *Dh44*, *Dh44R1*, and *Dh44R2* mutants and their roles in lipid accumulation. (**A**) Total triacylglycerol (TAG) level (µg TAG/mg fly) measurement in whole body extracts from control (*w^1118^*), *Dh44^Mi^*, *Dh44R1^Mi^*, and *Dh44R2^Mi^* mutants (n=5). (**B-E**) Nile red stains of the fat body from *w^1118^*, *Dh44^Mi^*, *Dh44R1^Mi^*, and *Dh44R2^Mi^*, respectively. Scale bars represent 50 µm. (**F**) Area of lipid droplets (LDs) in each indicated genotype (n=3). (**G**) The protein (µg protein/mg fly) measurement in the whole-body extracts from control (*w^1118^*) and *Dh44^Mi^*, *Dh44R1^Mi^*, and *Dh44R2^Mi^* mutants (n=4). (**H**) Quantification (qRT-PCR) of lipolytic gene (bmm) expression level in the fat body of *w^1118^*, *Dh44^Mi^*, *Dh44R1^Mi^*, and *D44R2^Mi^* flies under sated and starved conditions (n=3). (**I**) Total TAG level measurement in whole body extracts from the indicated genotypes (n=4). (**J-M**) Nile red stains of the fat body from (**J**) *w^1118^*, (**K**) *trpγ^1^* (**L**) *trpγ^1^,UAS-trpγ^1^/trpγ^1^,Dh44R1-GAL4*, and (**M**) *trpγ^1^,UAS-trpγ^1^/trpγ^1^,Dh44R2-GAL4*. (**N**) Area of LDs in each indicated

*Figure 7 continued on next page*

*Figure 7 continued*

genotype (n=3). Scale bars represent 50 μm. (**O–Q**) Immunohistochemistry with anti-GFP and anti-DsRed. (**O, P**) Co-expression of *trpγ* and *Dh44R2* (*trpγ*^G4/Dh44R2-LexA;UAS-mCD8::GFP/LexAopmCherry) in the R2 region of the intestine. Arrows indicate co-expression of *trpγ* and *Dh44R2* in the gut cell. (**O**) Full gut image. Scale bar represents 300 μm. (**P**) Magnified view of boxed R2 region in (**O**). Scale bar represents 50 μm. (**Q**) Co-expression of *Dh44R2* and *Dh44* (*Dh44R2-GAL4/Dh44-LexA;UAS-mCD8::GFP/LexAop-mCherry*) in the brain. Arrows indicate coexpressed two cells in the PI. Scale bar represents 50 μm. Means ± SEMs. Single-factor ANOVA with Scheffe's analysis was used as a post hoc test to compare multiple sets of data. The asterisks indicate significance from control (*p<0.05, **p<0.01). Each dot indicates distribution of individual sample value.

The online version of this article includes the following figure supplement(s) for figure 7:

**Figure supplement 1.** Measurement of starvation survival time with feeding lipase, glycerol, hexanoic acid, and triglyceride mix in 1% agar food.

flies exhibited lower carbohydrate levels and increased lipid levels. We previously proposed that TRPγ holds DH44 neurons in a state of afterdepolarization, thus reducing firing rates by inactivating voltage-gated Na⁺ channels (***Dhakal et al., 2022***). At the physiological level, this induces the consistent release of DH44 and depletion of DH44 stores, resulting in nutrient utilization and storage malfunctions. Likewise, our findings revealed that *Dh44* mutant and its receptor mutant, *Dh44R2*, displayed defects analogous to those observed in the *trpγ* mutant. The *trpγ* mutant phenotype in lipid regulation can be restored by the expression of *trpγ* in the *Dh44-GAL4* as well as *Dh44R2-GAL4*. This indicates that TRPγ functions in the brain and gut independently for lipid homeostasis. The mode of action for TRPγ in the gut should be further investigated. Finally, it would also be interesting to investigate the potential roles of TRPC members in humans, especially the involvement of TRPC4 and TRPC5 in metabolic syndrome.

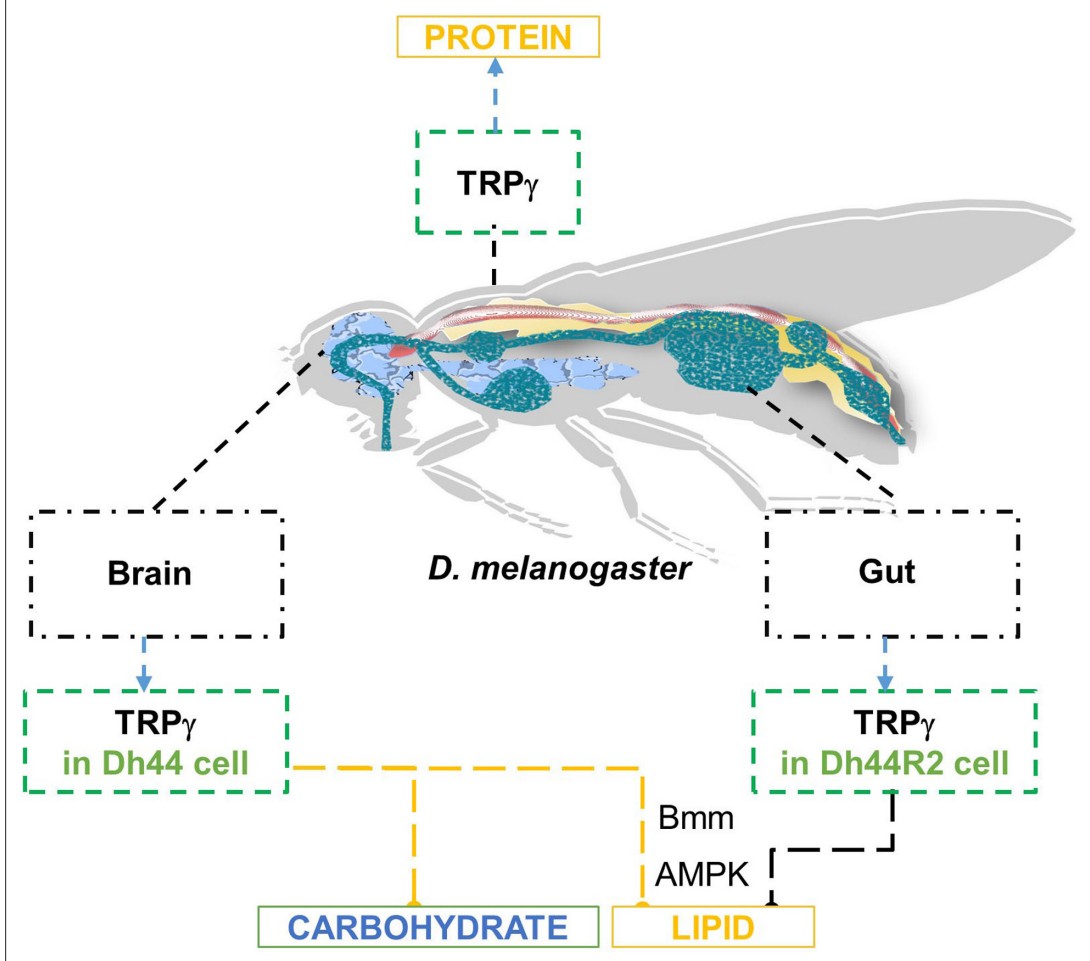

**Figure 8.** Schematic model representing the roles of *trpγ* in metabolic regualtion of carbohydrate, lipid, and protein.

Multiple lines of evidence have suggested that DH44 neurons communicate with the intestine. First, DH44 at the PI in the brain is an incretin-like hormone that acts as a hemolymph nutrient sensor. The secretion of DH44 is highly dependent on the calorie value of a given diet, as DH44 secretion is only mediated by nutritional sugars but not non-nutritional sugars (*Dhakal et al., 2022*; *Dus et al., 2015*; *Oh et al., 2021*). Second, the expression of TRPγ in the PI is sufficient to recover any metabolic defects such as reduced carbohydrate and increased lipid levels, as well as related phenomena such as starvation resistance. DH44 and kinin neuropeptides influence desiccation and starvation tolerance in *D. melanogaster* (*Cannell et al., 2016*). There is an inconsistency between total TAG levels and the LD size observed in the *Dh44* mutant. This inconsistency raises a possibility that the signaling pathway from DH44 release to its receptor DH44R2 only accounts for part of the lipid metabolic process under starvation. While *Dh44* mutant flies displayed normal internal TAG levels, *Dh44R2* mutant flies exhibited elevated TAG levels. This suggested that the lipolysis phenotype could be facilitated by a neuropeptide other than DH44. Alternatively, a DH44 neuropeptide-independent pathway could mediate the lipolysis. Although *Dh44^Mi* exhibited normal TAG levels, the presence of larger LDs indicated that DH44 plays a regulatory role in lipid metabolism. LDs consist of a hydrophobic core of neutral storage fats, such as triglycerides or cholesterol esters, encased by a protein-coated phospholipid monolayer. From our findings, it is suggested that DH44 and its receptor DH44R2, but not DH44R1, are involved in the control of lipid storage, indicating a specific pathway through which DH44 influences lipid metabolism distinct from its role in carbohydrate metabolism and overall energy homeostasis. TRPγ in DH44 neurons may influence the release of metabolic signals or hormones that act on postsynaptic DH44R2 cells. These postsynaptic cells could, in turn, modulate lipid storage and metabolism in a non-cell autonomous manner. However, the mechanism by which TRPγ functions in DH44R2 cells remains unclear. One possible explanation is that TRPγ in the gut may be activated by stretch or osmolarity (*Akitake et al., 2015*). Third, *trpγ* gene knockdown through RNAi in the DH44 neurons is enough to phenocopy all of the metabolic defects observed in the *trpγ* mutants. Fourth, *trpγ* mutants exhibit low expression of *brummer* lipase, which results in the accumulation of LDs in the body, which is a reciprocal process regulated by DH44 neurons. Brummer lipase is essential for regulating lipid levels in the insect fat body by mediating lipid mobilization and energy homeostasis. In *Nilaparvata lugens*, it facilitates triglyceride breakdown (*Lu et al., 2018*), while studies in *Drosophila* show that reduced Brummer lipase expression decreases fatty acids and increases diacylglycerol levels, highlighting its role in lipid metabolism (*Nazario-Yepiz et al., 2021*). Here, we additionally demonstrate that *bmm* expression in DH44 neurons within the PI region can systemically regulate TAG levels. Cell signaling or energy status in DH44 neurons may contribute to hormonal release that targets organs such as the fat body. Fifth, the activation of AMPK in the DH44 neurons can recover the defects of the lipid level of *trpγ^1*. Sixth, the reduced starvation resistance can be significantly recovered through dietary lipase supplementation. These findings support the assertion that the sugar-sensing DH44 neurons send downward signals to the intestine by absorbing the carbohydrates and lipids derived from dietary intake. However, DH44 neurons do not regulate protein levels in the body because the increased protein levels of *trpγ* mutants were not recovered by the expression of *trpγ* in the DH44 neurons. These findings confirm that *trpγ* regulates *bmm* expression in the adult bowel, enabling dietary lipid digestion and maintaining systemic lipid homeostasis.

Treatment with metformin, a lipolytic agent, at two different concentrations (1 mM and 5 mM) decreased the total TAG levels and reduced the size of the LDs in *D. melanogaster*. Similarly, metformin increased the starvation resistance of the *trpγ^1* mutant but not the controls. These findings were consistent with the widely acknowledged anti-obesity properties of metformin (*Sang et al., 2021*). Furthermore, our results were in agreement with the evidence that metformin helps to reduce lipid accumulation caused by diets rich in saturated fatty acids (*Kim et al., 2020*). Although the exact target of metformin is not known, its lipolytic effect is significant and widely documented. Oral administration of 5 mM of metformin reduced normal TAG levels of wild-type flies, whereas the 1 mM dose had no significant effect. Therefore, low metformin doses were effective at decreasing stored lipid levels in flies. The accumulation of intracellular LDs is regulated by autophagy. Metformin improves autophagic flux through AMPKα-mediated mechanisms (*Kim et al., 2020*). In the *Drosophila* context, we propose a mechanism involving AMPK activation within DH44 neurons as a rescue mechanism for starvation resistance in *trpγ^1*, highlighting its crucial role in this process. AMPK activation is known to enhance an organism's ability to endure environmental stresses by facilitating autophagy-mediated

cellular cleanup, promoting mitochondrial biogenesis for increased energy production, and optimizing metabolic pathways for enhanced energy efficiency (*Kim et al., 2020*). While the precise nature of the connection between TRPγ, DH44 neuronal activity, and AMPK activation remains unspecified, it is conceivable that TRPγ influences AMPK activation or regulation within DH44 neurons, contributing to stress responses, ATP supply, and metabolic adaptations. Metformin can influence lipid metabolism through various mechanisms, one of which is the promotion of lipolysis (*Szkudelski et al., 2022*). This effect results in a gradual reduction of stored lipid levels over time. Orally administering metformin to flies led to a significant reduction in lipid levels, accompanied by an increase in survival. Subsequent qPCR analysis post-metformin feeding revealed a significant upregulation in the expression level of the lipolytic gene (*bmm*). These results strongly support our hypothesis that metformin induces lipolysis when orally administered.

The exact mechanism through which orally administered lipase leads to gastric lipolysis remains unclear. However, our experimental data indicated that 0.1% lipase increased the survival time of *trpγ[1]* under starvation. Lipase exhibits a high degree of chemical selectivity and has the ability to catalyze triglyceride into glycerol ester, monoglycerides, glycerol, and fatty acids (*Liu et al., 2020*). Free fatty acids, HA, and glycerol at concentrations of 0.2–0.5% can be utilized as energy sources and increase survival time under starvation conditions, suggesting that *trpγ[1]* participates in the utilization of HA and glycerol under starvation conditions.

The therapeutic properties of metformin as a lipolytic and anti-diabetic drug have been widely documented in humans. However, additional studies are needed to identify the exact target of metformin. In this context, our findings demonstrate that *D. melanogaster* can be an excellent model to study the mode of action and biosafety of metformin.

## Materials and methods

### Key resources table

| Reagent type (species) or resource | Designation | Source or reference | Identifiers | Additional information |
|---|---|---|---|---|
| Genetic reagent (*Drosophila melanogaster*) | *trpγ[1]* | Bloomington *Drosophila* Stock Center | BDSC:64311 | Provided by Dr. C. Montel |
| Genetic reagent (*D. melanogaster*) | *trpγ[G4]* | Bloomington *Drosophila* Stock Center | BDSC:64313 | Provided by Dr. C. Montel |
| Genetic reagent (*D. melanogaster*) | *trpγ[1],UAStrpγ/CyO* | *Akitake et al., 2015* | | Provided by Dr. C. Montel |
| Genetic reagent (*D. melanogaster*) | *trpγ[1];g(trpγ)* | *Akitake et al., 2015* | | Provided by Dr. C. Montel |
| Genetic reagent (*D. melanogaster*) | *trpA1[1]* | Bloomington *Drosophila* Stock Center | BDSC:26504 | Provided by Dr. C. Montel |
| Genetic reagent (*D. melanogaster*) | *amo[1]* | *Watnick et al., 2003* | | Provided by Dr. C. Montel |
| Genetic reagent (*D. melanogaster*) | *trpml[2]* | Bloomington *Drosophila* Stock Center | BDSC:42230 | Provided by Dr. C. Montel |
| Genetic reagent (*D. melanogaster*) | *trp[343]* | Bloomington *Drosophila* Stock Center | BDSC:25082 | Provided by Dr. C. Montel |
| Genetic reagent (*D. melanogaster*) | *pyx[3]* | *Lee et al., 2005* | | Dr. Y. Lee |
| Genetic reagent (*D. melanogaster*) | *wtrw[ex]* | Bloomington *Drosophila* Stock Center | BDSC:59038 | Provided by Dr. C. Montel |
| Genetic reagent (*D. melanogaster*) | *trpl[29134]* | Bloomington *Drosophila* Stock center | BDSC:29134 | |
| Genetic reagent (*D. melanogaster*) | *iav[3621]* | Bloomington *Drosophila* Stock center | BDSC:24768 | |
| Genetic reagent (*D. melanogaster*) | *pain[2]* | *Tracey et al., 2003* | | Provided by Dr. S. Benzer |

*Continued on next page*

*Continued*

| Reagent type (species) or resource | Designation | Source or reference | Identifiers | Additional information |
|---|---|---|---|---|
| Genetic reagent (*D. melanogaster*) | nan$^{36a}$ | *Kim et al., 2003* | | Provided by Dr. C. Kim |
| Genetic reagent (*D. melanogaster*) | diLP2-GAL4 | Korea *Drosophila* Resource Center | KDRC: 200 | |
| Genetic reagent (*D. melanogaster*) | Dh44-GAL4 | Korea *Drosophila* Resource Center | KDRC: 2543 | Provided by Dr. Y. Kim |
| Genetic reagent (*D. melanogaster*) | UAS-mCD8::GFP | Bloomington *Drosophila* Stock Center | BDSC: 5130 | |
| Genetic reagent (*D. melanogaster*) | UAS-Kir2.1 | Bloomington *Drosophila* Stock Center | BDSC: 6596 | |
| Genetic reagent (*D. melanogaster*) | UAS-trpγ$^{RNAi}$ | Vienna *Drosophila* Resource center | Transformant ID107656 | |
| Genetic reagent (*D. melanogaster*) | r4-GAL4 | Korea *Drosophila* Resource Center | KDRC: 2166 | |
| Genetic reagent (*D. melanogaster*) | cg-GAL4 | Bloomington *Drosophila* Stock Center | BDSC: 7011 | Provided by Dr. S. Hyun |
| Genetic reagent (*D. melanogaster*) | Myo1A-GAL4 | Bloomington *Drosophila* Stock Center | BDSC: 67057 | Provided by Dr. S. Hyun |
| Genetic reagent (*D. melanogaster*) | UAS-AMPK$^{TD}$ | Korea *Drosophila* Resource Center | KDRC:10099 | |
| Genetic reagent (*D. melanogaster*) | UAS-bmm | Bloomington *Drosophila* Stock Center | BDSC: 76600 | |
| Genetic reagent (*D. melanogaster*) | Dh44$^{Mi}$ | Bloomington *Drosophila* Stock Center | BDSC: 24345 | Provided by Dr. Y. Kim |
| Genetic reagent (*D. melanogaster*) | Dh44R1$^{Mi}$ | Bloomington *Drosophila* Stock Center | BDSC: 23517 | Provided by Dr. Y. Kim |
| Genetic reagent (*D. melanogaster*) | Dh44R2$^{Mi}$ | Bloomington *Drosophila* Stock Center | BDSC: 29129 | Provided by Dr. Y. Kim |
| Genetic reagent (*D. melanogaster*) | Dh44R1-GAL4 | Korea *Drosophila* Resource Center | KDRC: 2734 | Provided by Dr. Y. Kim |
| Genetic reagent (*D. melanogaster*) | Dh44R2-GAL4 | Bloomington *Drosophila* Stock Center | BDSC: 66865 | Provided by Dr. Y. Kim |
| Genetic reagent (*D. melanogaster*) | Dh44R2$^{RNAi}$ | Korea *Drosophila* resource center | KDRC: 5121 | |
| Genetic reagent (*D. melanogaster*) | yw;+;bmm::GFP | *Men et al., 2016* | | Provided by Dr. Kaeko Kamei |
| Genetic reagent (*D. melanogaster*) | Dh44-LexA | Korea *Drosophila* Resource Center | KDRC: 2776 | |
| Genetic reagent (*D. melanogaster*) | Dh44R2-LexA | Korea *Drosophila* Resource Center | KDRC: 6616 | |
| Genetic reagent (*D. melanogaster*) | LexAop-mCherry | Korea *Drosophila* Resource Center | KDRC: 1247 | |
| Antibody | Mouse anti-GFP (mouse monoclonal) | Molecular probe | Cat # A11120 RRID:AB_221568 | 1:1000 (1 μL) |
| Antibody | Rabbit anti-DsRed(rabbit polyclonal) | Takara | Cat # 632496 RRID:AB_ 10013483 | 1:1000 (1 μL) |
| Antibody | Goat anti-mouse Alexa Fluor 488 | Thermo Fisher/ Invitrogen | Cat # A11029 RRID:AB_2534088 | 1:200 (1 μL) |

*Continued on next page*

*Continued*

| Reagent type (species) or resource | Designation | Source or reference | Identifiers | Additional information |
|---|---|---|---|---|
| Antibody | Goat anti-mouse Alexa Fluor 568 | Thermo Fisher/ Invitrogen | Cat # A11004 RRID:AB_2534072 | 1:200 (1 μL) |
| Antibody | Goat anti-rabbit Alexa Fluor 488 | Thermo Fisher/ Invitrogen | Cat # A11034 RRID:AB_2576217 | 1:200 (1 μL) |
| Antibody | Goat anti-rabbit Alexa Fluor 568 | Thermo Fisher/ Invitrogen | Cat # A11036 RRID:AB_10563566 | 1:200 (1 μL) |
| Antibody | Rabbit anti-Dh44 (rabbit polyclonal) | | | 1:500 Provided by Dr.J.A. Veenstra |
| Commercial assay or kit | Glucose (HK) Assay Kit | Sigma-Aldrich | Cat # GAHK-20 | |
| Commercial assay or kit | Glucose (HK) Assay reagent | Sigma-Aldrich | Cat # G3293 | |
| Commercial assay or kit | The Pierce BCA protein assay kit | Thermo Fischer Scientific | Cat # 23225 | |
| Commercial assay or kit | LiquiColor triglyceride test kit | Stanbio Laboratory | Cat # 2100–225 | |
| Chemical compounds or drug | Trehalase from porcine kidney | Sigma-Aldrich | Cat # 9025-52-9 | |
| Chemical compound or drug | Amyloglucosidase from Aspergillus | Sigma-Aldrich | Cat # 9032-08-0 | |
| Chemical compound or drug | Triglyceride mix | Sigma-Aldrich | Cat #17810-1AMP-S | |
| Chemical compound or drug | Lipase from *Candida rugosa* | Sigma-Aldrich | Cat # L1754 | |
| Chemical compound or drug | Metformin | Sigma-Aldrich | Cat # D150959 | |
| Chemical compound or drug | Hexanoic acid | Sigma-Aldric | Cat # W255912 | |
| Chemical compound or drug | Sucrose | Sigma-Aldrich | Cat # 57-50-1 | |
| Chemical compound or drug | Glycerol | Sigma-Aldrich | Cat # G5516 | |
| Chemical compound or drug | Nile red | Sigma-Aldrich | Cat # N3013 | |
| Software, algorithm | Origin Pro Version | OriginLab corporation | RRID:SCR_002815 | https://www.originlab.com/ |
| Software, algorithm | Graphpad Prism | GraphPad | RRID:SCR_002798 | https://www.graphpd.com/ |
| Software, algorithm | Fiji(ImageJ-win64) | Fiji | RRID:SCR_002285 | https://fiji.sc |

## Chemical sources

Trehalase from porcine kidney (cat. # 9025-52-9), amyloglucosidase (cat. # 9032-08-0), the glucose (HK) assay kit (cat. # GAHK-20-1KT), and triglyceride mix (cat. # 17810-1AMP-S), lipase from *Candida rugosa* (cat. # OT705690000), metformin (cat. # D150959), Nile Red (cat. # N3013), glycerol (cat. # G5516) were purchased from Sigma-Aldrich Co. Hexanoic acid (cat. # W255912) were purchased from Wako Pure Chemical Industry Ltd. The Pierce BCA protein assay kit (cat. # 23225) and the LiquiColor triglyceride test kit (cat. # 2100–225) were purchased from Thermo Fisher Scientific and Stanbio Laboratory, respectively.

## TAG level measurement

TAG levels were quantified as described previously with some modifications using a LiquiColor Triglyceride Test kit (cat. # 2100–225, Stanbio Laboratory, Germany) (*Sang et al., 2021*). Ten 5–10- d-old male flies were weighed and crushed in 1 mL of PBST (1 X PBS and 0.2 percent Triton X-100). The homogenate was incubated at 70 °C for 5 min and centrifuged for 3 min at 9500 g. Next, 100 μL of supernatant was transferred into a 1.5 mL Eppendorf tube and mixed with 1 mL of Stanbio LiquiColor Triglyceride Test kit reagent or 1 mL of deionized water to provide a baseline. The reaction mixture was kept at 37 °C for 15 min. Finally, the absorbance of the sample solution was measured at 500 nm using a spectrophotometer and the TAG level was calculated based on a standard calibration curve.

## Protein level measurement

Protein assays were performed as previously described using the Pierce BCA Protein Assay Kit with some modifications. Briefly, ten 5–10- d-old male flies were weighed and crushed in 1 mL of PBST (1 X PBS and 0.2 percent Triton X-10) and incubated at 70 °C for 5 min. The homogenate was then centrifuged for 3 min at 9500 g, after which 300 μL of supernatant was mixed with 600 μL of Pierce BCA Protein Assay Kit (UF289330). After a 30 min incubation period at 37 °C, the absorbance of the samples was measured at 530 nm using a spectrophotometer and compared to a standard calibration curve for quantification.

## Trehalose and glucose measurements in whole adult flies or hemolymph

Whole-body glucose and trehalose levels were measured in adult flies as previously described (*Meunier et al., 2007*). Briefly, ten 5–10- d old males were collected, weighed, and crushed in 250 μL of 0.25 M $Na_2CO_3$ buffer. The homogenates were then incubated in a water bath (95 °C) for 5 min to inactivate all enzymes. Next, 600 μL of 0.25 M sodium acetate and 150 μL of 1 M acetic acid (pH 5.2) were added to the samples, after which the mixtures were centrifuged at 12,500 g for 10 min at 24 °C. Afterward, 200 μL of supernatant was transferred to a new microfuge tube and 2 μL porcine kidney trehalase (Sigma: T8778 UN) was added to the sample mixture and incubated overnight at 37 °C to convert trehalose into glucose. Next, 1 mL of glucose hexokinase solution (Sigma: GAHK-20) was added to 100 μL of the sample and incubated for 20 min at 37 °C. Optical density (OD) values were measured at 340 nm. Finally, total glucose and trehalose levels were calculated using a standard glucose curve generated through similar reactions with standard trehalose and glucose.

Glucose and trehalose level measurements in hemolymph were extracted from the male flies. Flies were punctured in the thorax with the help of a fine injection needle and kept in 0.5 mL tubes having a punctured base with a 21-gauge needle. The punctured flies were adjusted by keeping the shoulder down to prevent leakage from the genital tract. The tubes were set into the 1.5 mL microfuge tube and centrifuged at 4 °C for 5 min at 2800 g force. 0.5 μL of hemolymph was added to 14.5 μL PBS. 2 μL of porcine kidney trehalase (Sigma: T8778 UN) was added to the sample mixture and incubated overnight at 37 °C to convert trehalose into glucose. 1 mL of glucose hexokinase solution (Sigma: GAHK-20) was added to the 100 μL of the sample and incubated at 37 °C for 20 min. Optical density (OD) values were measured at 340 nm. Total glucose and trehalose levels were calculated using a standard glucose curve generated through similar reactions with standard trehalose and glucose.

## Glycogen measurements

Tissue glycogen levels from whole-body extracts of adult flies were quantified as previously described (*Dus et al., 2011*). Briefly, 5 males were homogenized in 100 μL of ice-chilled 1 X PBS after measuring the weight. The homogenates were kept at 70 °C for 5 min to inactivate all metabolic enzymes. The homogenates were then centrifuged at 12,500 g for 3 min at 4 °C, after which 20 μL of the supernatant was transferred to 1.5 mL microfuge tubes and diluted with 1 X PBS to a 1:3 ratio. An amyloglucosidase dilution was prepared by mixing 1.5 μL of amyloglucosidase (Sigma A1602) suspension in 998.5 μL 1 X PBS. Finally, a 20 μL aliquot of the diluted sample was added to 20 μL of the diluted amyloglucosidase solution, as well as to the glycogen standard. Both the glycogen standard and test samples were then incubated at 37 °C for 1 hr. A commercial glucose (HK) assay reagent (Sigma: G3293 VER) was used to measure total glucose at 340 nm. The glycogen level was quantified by comparing it with a standard curve plotted from standard glycogen samples.

## Survival assay

Survival experiments were conducted as previously described (*Lee et al., 2018b*). Normal cornmeal diets were used to conduct the normal survival assays. To measure survival under starvation conditions, twenty 3–4 d-old male flies were fed with 1% agar food supplemented with or without various concentrations of lipase (0.1%; active or denatured), glycerol (1%), hexanoic acid (0.2%, 0.5%), glycerol (1%), TAG mix (0.2%, 0.5%), or metformin (1 mM, 5 mM). Every 12 hr, the flies were monitored/counted and then transferred to fresh vials with the same food supply. The experiments were conducted until the food vials had been cleared.

## Immunohistochemistry

We performed immunohistochemistry as previously described (*Dhakal et al., 2022*). Briefly, to fix and block the specimens, we placed freshly dissected tissues (brain and intestine) into a well of 24-well

tissue culture plate (Costar Corp.) placed on ice, which contained 940 µL of fixing buffer (1 mM EGTA, 0.1 M Pipes pH 6.9, 2 mM MgSO₄, 1% TritonX-100, 150 mM NaCl). 60 µL formaldehyde (37%) was added to the wells and mixed instantly before the tissues were added. As many as tissues that were dissected within 15 min were fixed and then the samples were incubated for 30 min more. Samples were washed with wash buffer (1 X PBS, 0.1% saponin) three times (15 min. for each washing), and blocked with 1 mL blocking buffer (1 X PBS, 0.1% saponin, and 5 mg/mL BSA) at 4 °C for 4–8 hr.

To perform immunostaining, primary antibodies were inserted into the sample at 4 °C for 18 hr (*Jais and Brüning, 2022*). Samples were washed with wash buffer three times (15 min each) and added secondary antibodies {(1:200) goat anti-mouse Alexa Fluor 488 (cat. # A11029), goat anti-mouse Alexa Fluor 568 (cat. # A11004), goat anti-rabbit Alexa Fluor 488 (cat. # A11034), and goat anti-rabbit Alexa 568 (cat. # A11036) at 4 °C for 4 hr. Finally, samples were washed three times and stored into 1.25 X PDA (187.5 mM NaCl, 37.5% glycerol, 62.5 mM Tris pH 8.8), and kept at 4 °C for more than 1 hr. Samples were mounted and examined using a Leica Stellaris 5 Confocal Microscope.

### Nile red staining

Nile Red is a dark purplish-red powder (Sigma N-3013), the stock solution must be prepared in acetone (1000 µg/mL) and kept in a tightly sealed, lightproof container at 4 °C. Briefly, 5–10 d-old male flies were fixed in a sagittal position on a glass slide and submerged in a 1 X PBS solution. FBs were gently dissected from a dorsal abdominal region along with a thorax or the gut under a stereomicroscope. The dissected tissues were fixed with 4% formaldehyde solution for 15 min at room temperature. The fixed tissues were gently washed three times with 1 X PBS (5 min for each wash). Nile Red (1:1000 dilution) was added to the tissue samples for 5 min. Finally, the stained tissues were washed with 1 mL 1 X PBS and mounted in 50% glycerol on a glass slide. LD deposition exhibits greater density in the upper abdominal region compared to the lower abdominal area. This comparison was meticulously conducted within the identical segments 2–3 of the abdomen to ensure accuracy.

For intestinal lipid staining, 5–10 d-old male flies were fixed on a glass slide and submerged in 1 X PBS solution. Full intestine was dissected very carefully. The dissected intestine was fixed with a 4% formaldehyde solution for 15 min at room temperature. The fixed intestine was washed 3 times with 1 X PBS. Nile red was added to the sample for 5 min. The stained samples were washed with 1 mL 1 X PBS and mounted in 50% glycerol on a glass slide. The stained tissues were examined using a Leica Stellaris 5 Confocal Microscope.

### qRT-PCR

qRT-PCR assays were performed as described previously (*Yoshinari et al., 2021*). Briefly, ten 6–10 d-old male flies (control and mutant) were selected for the whole body and 15 flies were selected for fat body samples in *Figure 7H*. The experiments were conducted under sated (0 hr starvation) and starved (24 hr starvation) conditions. Total RNA was extracted using the TRizol reagent (Invitrogen) followed by DNase (Promega) treatment. cDNA was synthesized using the AMV reverse transcriptase system (Promega). RT-qPCR experiments were carried out using a Bio-rad CFX system. The Takara TB Green Premix was used to assess the mRNA expression level of each gene according to the manufacturer's instructions. Relative gene expression was calculated using the $2^{-\Delta\Delta Ct}$ method. Three biological samples were used and transcript levels were normalized to the *D. melanogaster* housekeeping gene *tubulin*. The experiments were conducted using the following primer pairs: *acc*, 5'-ACG AGG GCG AGC AGC GTT AC-3' (forward) and 5'-TAG GGC GAC TTG GTG GGC AT-3' (reverse); *bmm*, 5'-ATG ACT TCG GAC TTC TTC AGG G-3' (forward) and 5'-CCA ATT CAG ATG GAA GAG CTG-3' (reverse); *fbp*, 5'-CTC CAA CGA GCT GTT CAT CA-3' (forward) and 5'-TGA ACC GAT CGA CAC CAG GC-3' (reverse); *pepck1*, 5'-AGG TGC ACA TCT GCG ATG GC-3' (forward) and 5'-CCA CCA CGT AAG CAG AGT CC-3' (reverse); *desat1*, 5'-AAG CCG GTG CCC AGT CCA TC-3' (forward) and 5'-ATG GTC GCG AGC CCA ATG GT-3' (reverse); and *tubulin*, 5'-TCC TTG TCG CGT GTG AAA CA-3' (forward) and 5'-CCG AAC GAG TGG AAG ATG AG-3' (reverse).

### Statistics and reproducibility

*D. melanogaster* was selected as a model organism in this study. For the experiments, male flies were mostly used unless we did not mention the sex. All of the experiments were conducted under laboratory conditions. The appropriate number of replicates was established based on previous research. A

large enough sample size was used in all of our assays to ensure that our results were representative and repeatable. No data points were left out of the analysis. For each genotype, the data points indicate the values of individual replicates. The error bars in all of the figures represent the standard error of the mean (SEM). The analysis of the RT-qPCR data was conducted using the $C_T$ values. Comparisons between multiple experimental groups were conducted via single-factor ANOVA and Scheffe's *post hoc* test. Pair-wise comparisons were conducted via Student's t-test. Survival curves were estimated for each group, using a Kaplan-Meier method and compared statistically using the log-rank tests. The asterisks in the figures indicate statistical significance (*p<0.05, **p<0.01). All statistical analyses were conducted using the Origin Pro 8 software for Windows (ver. 8.0932; Origin Lab Corporation, USA).

## Acknowledgements

This work was supported by grants to Youngseok Lee from the National Research Foundation of Korea (NRF) funded by the Korea government (MIST) (RS-2021-NR058319) and the Biomaterials Specialized Graduate Program through the Korea Environmental Industry & Technology Institute (KEITI) funded by the Ministry of Environment (MOE). DKN and SD were supported by the Global Scholarship Program for Foreign Graduate Students at Kookmin University in Korea.

## Additional information

### Funding

| Funder | Grant reference number | Author |
| --- | --- | --- |
| National Research Foundation of Korea | RS-2021-NR058319 | Youngseok Lee |
| Kookmin University | Global Scholarship Program for Foreign Graduate Students | Dharmendra Kumar Nath Subash Dhakal |
| Korea Environmental Industry and Technology Institute | Biomaterials Specialized Graduate Program | Youngseok Lee |

The funders had no role in study design, data collection and interpretation, or the decision to submit the work for publication.

### Author contributions

Dharmendra Kumar Nath, Data curation, Validation, Investigation, Visualization, Methodology, Writing – original draft; Subash Dhakal, Conceptualization, Investigation, Methodology, Writing – original draft; Youngseok Lee, Conceptualization, Supervision, Funding acquisition, Writing – review and editing

### Author ORCIDs

Dharmendra Kumar Nath ⬤ https://orcid.org/0000-0001-6105-6970
Youngseok Lee ⬤ https://orcid.org/0000-0003-0459-1138

Reviewer #1 (Public review): https://doi.org/10.7554/eLife.99258.3.sa1
Reviewer #2 (Public review): https://doi.org/10.7554/eLife.99258.3.sa2
Reviewer #3 (Public review): https://doi.org/10.7554/eLife.99258.3.sa3
Author response https://doi.org/10.7554/eLife.99258.3.sa4

## Additional files

### Supplementary files
MDAR checklist

## Data availability

Source data for all figures contained in the manuscript have been deposited in figshare.

The following dataset was generated:

| Author(s) | Year | Dataset title | Dataset URL | Database and Identifier |
|---|---|---|---|---|
| Nath DK, Dhakal S, Lee Y | 2024 | TRPγ regulates lipid metabolism through Dh44 neuroendocrine cells | https://doi.org/10.6084/m9.figshare.28029974 | figshare, 10.6084/m9.figshare.28029974 |

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
