## [Editor Report · eLife Assessment]

This **important** study reports findings that Trpγ, a type of transient receptor potential (TRP) channel expressed in Dh44-releasing neuroendocrine cells, mediates starvation-dependent lipid catabolism. Overall, the claims of the authors are supported by **solid** evidence. The work should be of interest to both basic and medical biologists working on lipid metabolism.

---

## [Referee Report · Reviewer #1 (Public review)]

Summary:

This research article by Nath et al. from the Lee Lab addresses how lipolysis under starvation is achieved by a transient receptor potential channel, TRPγ, in the neuroendocrine neurons to help animals survive prolonged starvation. Through a series of genetic analyses, the authors identify that trpγ mutations specifically lead to a failure in lipolytic processes under starvation, thereby reducing animals' starvation resistance. The conclusion was confirmed through total triacylglycerol levels in the animals and lipid droplet staining in the fat bodies. This study highlights the importance of transient receptor potential (TRP) channels in the fly brain to modulate energy homeostasis and combat metabolic stress. However, the co-expression of trpγ and Dh44-R2 in the gut is not convincing, especially in the picture of the arrows pointing at the autofluorescence signals in the gut (Figure 7P). Therefore, the authors should either confirm the co-expression or acknowledge that trpγ and Dh44-R2 are not co-expressed in the gut and modify their model in Figure 8 accordingly, although clarifying their co-expression may not change the main conclusions of this study. Overall, the revised version includes the required clarifications on their important results that strengthen the interpretations of the research as well as the visibility of this study.

Strengths:

This study identifies the biological meaning of TRPγ in promoting lipolysis during starvation, advancing our knowledge about the TRPγ channel and the neural mechanisms to combat metabolic stress. Furthermore, this study demonstrates the potential of the TRPγ channel as a target to develop new therapeutic strategies for human metabolic disorders by showing that metformin and AMPK pathways are involved in its function in lipid metabolisms during starvation in *Drosophila.*

---

## [Referee Report · Reviewer #2 (Public review)]

Summary

In this paper, the function of trpγ in lipid metabolism was investigated. The authors found that lipid accumulation levels were increased in trpγ mutants and remained high during starvation; the increased TAG levels in trpγ mutants were restored by the expression of active AMPK in DH44 neurons and oral administration of the anti-diabetic drug metformin. Furthermore, oral administration of lipase, TAG and free fatty acids effectively restored survival of trpγ mutants under starvation conditions. These results indicate that TRPv plays an important role in the maintenance of systemic lipid levels through the proper expression of lipase. Furthermore, authors have shown that this function is mediated by DH44R2. This study provides an interesting finding in that the neuropeptide DH44 released from the brain regulates lipid metabolism through a brain-gut axis, acting on the receptor DH44R2 expressed in gut cells.

Strengths

Using *Drosophila* genetics, careful analysis of which cells express trpγ regulates lipid metabolism is performed in this study. The study supports its conclusions from various angles, including not only TAG levels, but also fat droplet staining and survival rate under starved conditions, and oral administration of substances involved in lipid metabolism.

Weaknesses

The function of lipases, as well as identification of cell types, in the DH44R2-expressing cells in the gut can be investigated.

---

## [Referee Report · Reviewer #3 (Public review)]

In this manuscript, the authors demonstrated the significance of the TRPγ channel in regulating internal TAG levels. They found high TAG levels in TRPγ mutant, which was ascribed to a deficit in the lipolysis process due to the downregulation of brummer (bmm). It was notable that the expression of TRPγ in DH44+ PI neurons, but not dILP2+ neurons, in the brain restored the internal TAG levels and that the knockdown of TRPγ in DH44+ PI neurons resulted in an increase in TAG levels. These results suggested a non-cell autonomous effect of Dh44+PI neurons. Additionally, the expression of the TRPγ channel in Dh44 R2-expressing cells restored the internal TAG levels. The authors, however, did not provide an explanation of how TRPγ might function in both presynaptic and postsynaptic cells in the non-cell autonomous manner to regulate the TAG storage. The authors further determined the effect of TRPγ mutation on the size of lipid droplets (LD) and the lifespan and found that TRPγ mutation caused an increase in the size of LD and a decrease in the lifespan, which were reverted by feeding lipase and metformin. These were creative endeavors, I thought. The finding that DH44+ PI neurons have non-cell autonomous functions in regulating bodily metabolism (mainly sugar/lipid) in addition to directing sugar nutrient sensing and consumption is likely correct, but the paper has many loose ends.

Comments on revisions:

The authors have addressed nearly all of my concerns with additional experiments and explanations.

---

## [Author Response]

The following is the authors’ response to the original reviews

**Public Reviews:**

**Reviewer #1 (Public Review):**
Summary:This research article by Nath et al. from the Lee Lab addresses how lipolysis under starvation is achieved by a transient receptor potential channel, TRPγ, in the neuroendocrine neurons to help animals survive prolonged starvation. Through a series of genetic analyses, the authors identify that TRPγ mutations specifically lead to a failure in lipolytic processes under starvation, thereby reducing animals' starvation resistance. The conclusion was confirmed through total triacylglycerol levels in the animals and lipid droplet staining in the fat bodies. This study highlights the importance of transient receptor potential (TRP) channels in the fly brain to modulate energy homeostasis and combat metabolic stress. While the data is compelling and the message is easy to follow, several aspects require further clarification to improve the interpretation of the research and its visibility in the field.Strengths:This study identifies the biological meaning of TRPγ in promoting lipolysis during starvation, advancing our knowledge about TRP channels and the neural mechanisms to combat metabolic stress. Furthermore, this study demonstrates the potential of the TRP channel as a target to develop new therapeutic strategies for human metabolic disorders by showing that metformin and AMPK pathways are involved in its function in lipid metabolisms during starvation in *Drosophila*.Weaknesses:Some key results that might strengthen their conclusions were left out for discussion or careful explanation (see below). If the authors could improve the writing to address their findings and connect their findings with conclusions, the research would be much more appreciated and have a higher impact in the field.Here, I listed the major issues and suggestions for the authors to improve their manuscript:(1) Are the increased lipid droplet size and the upregulated total TAG level measured in the starved or sated mutant in Figure 1? This information might be crucial for readers to understand the physiological function of TRP in lipid metabolism. In other words, clarifying whether the upregulated lipid storage is observed only in the starved trp mutant will advance our knowledge of TRPγ. If the increase of total TAG level is only observed in the starved animals, TRP in the Dh44 neurons might serve as a sensor for the starvation state required to promote lipolysis in starvation conditions. On the other hand, if the total TAG level increases in both starved and sated animals, activation of Dh44 through TRPγ might be involved in the lipid metabolism process after food ingestion.

We measured total TAG level in Figure 1 and LD sizes in Figure 2 under sated condition. We inserted “under sated condition” to clarify it. lines 97 and 147-148.

Thanks for your suggestions.

(2) It is unclear how AMPK activation in Dh44 neurons reduces the total triacylglycerol (TAG) levels in the animals (Figure 3G). As AMPK is activated in response to metabolic stress, the result in Figure 3G might suggest that Dh44 neurons sense metabolic stress through AMPK activation to promote lipolysis in other tissues. Do Dh44 neurons become more active during starvation? Is activation of Dh44 neurons sufficient to activate AMPK in the Dh44 neurons without starvation? Is activation of AMPK in the Dh44 neurons required for Dh44 release and lipolysis during starvation? These answers would provide more insights into the conclusion in Lines 192-193.

In our previous study, we demonstrated that trpγ mutants exhibited lower levels of glucose, trehalose and glycogen level (Dhakal et al. 2022), and in the current study, we observed excessive lipid storage in the trpγ mutant, indicating imbalanced energy homeostasis. Given the established role of AMPK in maintaining energy balance (Marzano et. al., 2021, Lin et al 2021), we employed the activated form of AMPK (*UAS-AMPKTD*) in our experiments. Our result showed that expression of activated AMPK in *Dh44* neurons led to a reduction in total TAG levels, suggesting that AMPK activation in these neurons can promote lipolysis even in the absence of starvation. Regarding the activation of *Dh44* neurons, Dus et al in 2015 reported that *Dh44* cells in the brain are activated by nutritive sugars especially in starvation conditions. In addition, another report showed a role of Dh44 neuron in regulating starvation induced sleep suppression (Oh et. al., 2023) which may imply that these neurons become more active under starved conditions. We did not directly assess whether Dh44 neuron activity increases during starvation or whether AMPK activation in these neurons is required for DH44 release and subsequent lipolysis, our finding support the notion that AMPK activation in Dh44 neuron is sufficient to reduce TAG levels, potentially by metabolic stress response typically observed during starvation. We explained it like the following: “*Dh44* neurons regulate starvation-induced sleep suppression (Oh et. al., 2023), which implies that these neurons become more active under starved conditions.” lines 190-191.

(3) It is unclear how the lipolytic gene brummer is further downregulated in the trpγ mutant during starvation while brummer is upregulated in the control group (Figure 6A). This result implies that the trpγ mutant was able to sense the starvation state but responded abnormally by inhibiting the lipolytic process rather than promoting lipolysis, which makes it more susceptible to starvation (Figure 3B).

Thanks for your suggestions. We explained it like the following: “The data indicates that the *trpg* mutant can sense the starvation state but responds abnormally by suppressing lipolysis instead of activating it. This dysregulated lipolytic response likely increases the mutant's vulnerability to starvation, as it cannot effectively mobilize lipid stores for energy during periods of nutrient deprivation.” lines 251-254.

(4) There is an inconsistency of total TAG levels and the lipid droplet size observed in the Dh44 mutant but not in the Dh44-R2 mutant (Figures 7A and 7F). This inconsistency raises a possibility that the signaling pathway from Dh44 release to its receptor Dh44-R2 only accounts for part of the lipid metabolic process under starvation. Adding discussion to address this inconsistency may be helpful for readers to appreciate the finding.

Thanks for your suggestion. We included the following in the Discussion: “There is an inconsistency of total TAG levels and the LD size observed in the *Dh44* mutant. This inconsistency raises a possibility that the signaling pathway from DH44 release to its receptor DH44R2 only accounts for part of the lipid metabolic process under starvation. While *Dh44* mutant flies displayed normal internal TAG levels, *Dh44R2* mutant flies exhibited elevated TAG levels. This suggested that the lipolysis phenotype could be facilitated by a neuropeptide other than DH44. Alternatively, a DH44 neuropeptide-independent pathway could mediate the lipolysis.” lines 429-436.

**Reviewer #2 (Public Review):**
Summary:In this paper, the function of trpγ in lipid metabolism was investigated. The authors found that lipid accumulation levels were increased in trpγ mutants and remained high during starvation; the increased TAG levels in trpγ mutants were restored by the expression of active AMPK in DH44 neurons and oral administration of the anti-diabetic drug metformin. Furthermore, oral administration of lipase, TAG, and free fatty acids effectively restored the survival of trpγ mutants under starvation conditions. These results indicate that TRPv plays an important role in the maintenance of systemic lipid levels through the proper expression of lipase. Furthermore, authors have shown that this function is mediated by DH44R2. This study provides an interesting finding in that the neuropeptide DH44 released from the brain regulates lipid metabolism through a brain-gut axis, acting on the receptor DH44R2 presumably expressed in gut cells.Strengths:Using *Drosophila* genetics, careful analysis of which cells express trpγ regulates lipid metabolism is performed in this study. The study supports its conclusions from various angles, including not only TAG levels, but also fat droplet staining and survival rate under starved conditions, and oral administration of substances involved in lipid metabolism.Weaknesses:Lipid metabolism in the gut of DH44R2-expressing cells should be investigated for a better understanding of the mechanism. Fat accumulation in the gut is not mechanistically linked with fat accumulation in the fat body. The function of lipase in the gut (esp. R2 region) should be addressed, e.g. by manipulating gut-lipases such as magro or Lip3 in the gut in the contest of trpγ mutant. Also, it is not clarified which cell types in the gut DH44R2 is expressed. The study also mentioned only in the text that bmm expression in the gut cannot restore lipid droplet enlargement in the fat body, but this result might be presented as a figure.

We appreciate the reviewer’s insightful suggestions. Unfortunately, due to the unviability of the reagent (*UAS-Lip3*), we were unable to manipulate gut lipase in trpy mutants as proposed. However, we additionally performed immunostaining to examine the co-expression of trpγ and Dh44R2 in the gut, and our results indicate that both trpγ and Dh44R2 are co-expressed in the R2 region of the gut (Figure 7O and P). Furthermore, we have updated our figures to address the point that *bmm* expression in the gut does not restore lipid droplet enlargement in the fat body, with the revised version (Figure 5I and J).

**Reviewer #3 (Public Review):**
In this manuscript, the authors demonstrated the significance of the TRPγ channel in regulating internal TAG levels. They found high TAG levels in TRPγ mutant, which was ascribed to a deficit in the lipolysis process due to the downregulation of brummer (bmm). It was notable that the expression of TRPγ in DH44+ PI neurons, but not dILP2+ neurons, in the brain restored the internal TAG levels and that the knockdown of TRPγ in DH44+ PI neurons resulted in an increase in TAG levels. These results suggested a non-cell autonomous effect of Dh44+PI neurons. Additionally, the expression of the TRPγ channel in Dh44 R2-expressing cells restored the internal TAG levels. The authors, however, did not provide an explanation of how TRPγ might function in both presynaptic and postsynaptic cells in the non-cell autonomous manner to regulate the TAG storage. The authors further determined the effect of TRPγ mutation on the size of lipid droplets (LD) and the lifespan and found that TRPγ mutation caused an increase in the size of LD and a decrease in the lifespan, which were reverted by feeding lipase and metformin. These were creative endeavors, I thought. The finding that DH44+ PI neurons have non-cell autonomous functions in regulating bodily metabolism (mainly sugar/lipid) in addition to directing sugar nutrient sensing and consumption is likely correct, but the paper has many loose ends. I would like to see a revision that includes more experiments to tighten up the findings and appropriate interpretations of the results.(1) The authors need to provide interpretations or speculations as to how DH44+ PI neurons have non-cell autonomous functions in regulating the internal TAG stores, and how both presynaptic DH44 neurons and postsynaptic DH44 R2 neurons require TRPγ for lipid homeostasis.

In Discussion, we had mentioned our previous finding. “ We previously proposed that TRPg holds DH44 neurons in a state of afterdepolarization, thus reducing firing rates by inactivating voltage-gated Na+ channels (Dhakal et al., 2022). At the physiological level, this induces the consistent release of DH44 and depletion of DH44 stores, resulting in nutrient utilization and storage malfunctions.”

We also included the following: “TRPg in DH44 neurons may influence the release of metabolic signals or hormones that act on postsynaptic DH44R2 cells. These postsynaptic cells could, in turn, modulate lipid storage and metabolism in a non-cell autonomous manner. However, the mechanism by which TRPg functions in DH44R2 cells remains unclear. One possible explanation is that TRPg in the gut may be activated by stretch or osmolarity (Akitake et al. 2015).” lines 439-440.

This interaction between presynaptic and postsynaptic cells may ensure a coordinated response to metabolic changes and maintain lipid homeostasis. Thus, both Dh44-expressing and Dh44-R2-expressing cells are crucial for the proper functioning of TRPγ in regulating internal TAG levels and lipid storage.

(2) The expression of TRPγ solely in DH44 R2 neurons of TRPγ mutant flies restored the TAG phenotype, suggesting an important function mediated by TRPγ in DH44 R2 neurons. However, the authors did not document the endogenous expression of TRPγ in the DH44R2+ gut cells. This needs to be shown.

We appreciate the reviewer’s suggestion. To address this, we performed immunostaining to examine the expression of TRPγ in the DH44R2+ gut cells. Our results, as shown in Figure 7 O and P, confirm that TRPγ is co-expressed in the Dh44R2+ cells in the gut. We also found that Dh44R2 is expressed in the brain as well. We documented this part like the following: “Given that *Dh44R2* is predominantly expressed in the intestine, we performed immunostaining to examine whether *Dh44R2* co-localizes with *trpg* in gut cells. Our results confirmed that *Dh44R2* and *trpg* are co-expressed in intestinal cells (Figure 7O and P). Additionally, we analyzed *Dh44R2* expression in the brain and found that two *Dh44R2*-expressing cells are co-localized with *Dh44*-expressing cells in the PI region (Figure 7Q). To further delineate whether *Dh44R2*-mediated fat utilization is specific to the brain, gut, or fat body, we knocked down *Dh44R2RNAi* using *Dh44-GAL4*, *myo1A-GAL4*, and *cg-GAL4*, respectively (Figure 7–figure supplement 1E). Notably, knockdown of *Dh44R2* with *Myo1A-GAL4* resulted in elevated TAG levels, indicating that DH44R2 activity in lipid metabolism is specific to the gut.” lines 375-384.

(3) While Dh44 mutant flies displayed normal internal TAG levels, Dh44R2 mutant flies exhibited elevated TAG levels (Figure 7A). This suggested that the lipolysis phenotype could be facilitated by a neuropeptide other than Dh44. Alternatively, a Dh44 neuropeptide-independent pathway could mediate the lipolysis. In either case, an additional result is needed to substantiate either one of the hypotheses.

The Dh44 mutant flies exhibited normal TAG levels, whereas Dh44R2 mutant flies showed elevated TAG levels. However, when we examined the lipid droplets in the fat body, both Dh44 mutant and Dh44R2 mutant flies displayed larger lipid droplets, indicating a disruption in lipid metabolism. Additionally, we assessed starvation survival time and found that both Dh44 and Dh44R2 mutant flies exhibited reduced survival under starvation conditions compared to controls. Supplementation with lipase (Figure 7–figure supplement 1A), glycerol (Figure 7–figure supplement 1B), hexanoic acid (Figure 7–figure supplement 1C), and mixed TAGs (Figure 7–figure supplement 1D) improved starvation survival time, further supporting that the lipid metabolism pathway was impaired in both mutants. These observations highlight the role of Dh44 in regulating lipolysis. We included related Discussion: “There is an inconsistency of total TAG levels and the LD size observed in the *Dh44* mutant. This inconsistency raises a possibility that the signaling pathway from DH44 release to its receptor DH44R2 only accounts for part of the lipid metabolic process under starvation. While *Dh44* mutant flies displayed normal internal TAG levels, *Dh44R2* mutant flies exhibited elevated TAG levels. This suggested that the lipolysis phenotype could be facilitated by a neuropeptide other than DH44. Alternatively, a DH44 neuropeptide-independent pathway could mediate the lipolysis.” lines 429-436.

(4) While the authors observed an increased area of fat body lipid droplets (LD) in Dh44 mutant flies (Figure 7F), they did not specify the particular region of the fat body chosen for measuring the LD area.

We have chosen the 2-3 segment in the abdomen for all fat body images, which we already mentioned in Nile red staining in the Method section line 630-631.

(5) The LD area only accounts for TAG levels in the fat body, whereas TAG can be found in many other body parts, including the R2 area as demonstrated in Figure 5A-D using Nile red staining. As such, measuring the total internal TAG levels would provide a more accurate representation of TAG levels than the average fat body LD area.

We have measured total internal TAG level in whole body throughout the experiments (Figure 1F, 2C, 2E, 3C, 3G, 4A, 4B, 7A, 7I, and many Supplementary Figures) except *bmm* expression using GAL4/UAS system. Now we include this new data in (Figure 5–figure supplement 1) which is the same conclusion with LD analysis.

(6) In Figure 5F-I, the authors should perform the similar experiment with Dh44, Dh44R1, and Dh44R2 mutant flies.We did the experiments with *Dh44*, *Dh44R1*, and *Dh44R2* mutant flies and we found that *Dh44* and *Dh44R2* mutant flies showed reduced starvation survival time than control and which was increased after supplementation of lipase, glycerol, hexanoic acid and TAG (Figure 7– figure supplement 1A–D). lines 361-372.(7) The representative image in Figure 6B does not correspond to the GFP quantification results shown in Figure 6C. In trpr1;bmm::GFP flies, the GFP signal appears stronger in starved conditions than in satiated conditions.

We updated it with new images. We quantified GFP intensity level using image J and found that GFP intensity level was significantly lower in starved condition in *trpγ1;bmm::GFP* flies than sated condition.

(8) In Figure 6H-I, fat body-specific expression of bmm reversed the increased LD area in TRPγ mutants. The authors also showed that Dh44+PI neuron-specific expression of bmm yielded a similar result. The authors need to provide an interpretation as to how bmm acts in the fat body or DH44 neurons to regulate this.

We first inserted the following in results: “Furthermore, the expression of *bmm* in the fat body, as well as *Dh44* neurons in the PI region, can promote lipolysis at the systemic level.” lines 276-277.

Additionally, we discussed it in the Discussion: “Brummer lipase is essential for regulating lipid levels in the insect fat body by mediating lipid mobilization and energy homeostasis. In *Nilaparvata lugens*, it facilitates triglyceride breakdown (Lu et al., 2018), while studies in *Drosophila* show that reduced Brummer lipase expression decreases fatty acids and increases diacylglycerol levels, highlighting its role in lipid metabolism (Nazario-Yepiz et al., 2021). Here, we additionally demonstrate that *bmm* expression in DH44 neurons within the PI region can systemically regulate TAG levels. Cell signaling or energy status in DH44 neurons may contribute to hormonal release that targets organs such as the fat body.” lines 451-459.

(9) The authors should explain why the DH44 R1 mutant did not represent similar results as the wild type.

We added “In addition, *bmm* levels in *Dh44R1Mi* under starved condition did not increase as significantly as in the control. This suggests a unique role of DH44 and its receptors in regulating lipid metabolism and response to nutritional status in *Drosophila*.” lines 358-360.

(10) It would be good to have a schematic that represents the working model proposed in this manuscript.

We updated the schematic model in revised version (Figure 8).

**Recommendations for the authors:**

**Reviewing Editor (Recommendations For The Authors):**
This paper characterized the function of trpγ in Dh44-expressing PI neurons for lipid metabolism and lipolysis induced by prolonged starvation. The authors applied a series of lipolytic genetic manipulation and lipid/lipid metabolism supplements to rescue the trpγ deficits in lipolysis: the expression of active AMPK in the DH44-expressing PI neurons or brummer, a lipolytic gene, in the trpγ-expressing cells, and oral administration of the anti-diabetic drug metformin, lipase, TAG and free fatty acids. Despite this exhaustive characterization of the defective lipolysis in the trpγ mutants, there remain puzzles in inconsistent defects of Dh44 and DH44R2 in the total TAG levels and in the expression and functions of the receptor in the gut. Clarification of these points and other issues raised by the reviewers should improve the mechanisms of lipid metabolism through Dh44 signalling.
**Reviewer #1 (Recommendations For The Authors):**
(1) It might be worth introducing Dh44 in the introduction section as it is unclear to readers how the authors hypothesized the site-of-action of TRPγ in Dh44 neurons for lipid metabolism after reading the introduction.

We introduced the following: “We found that TRPg expression in *Dh44* neuroendocrine cells in the brain is critical for maintaining normal carbohydrate levels in tissues (Dhakal et al. 2022). Building on this, we hypothesized that TRPg in *Dh44* cells also regulates lipid and protein homeostasis.” lines 69-71.

(2) Providing a summary model in the end to integrate the present findings and their previous publication about TRPγ functions in *Drosophila* sugar selection would greatly help readers understand and appreciate the general role of TRPγ in balancing energy homeostasis.

We made a schematic model in Figure 8.

(3) Swapping the order of Figures 5 and 6 might be a better way to tell the story without logic gaps. The results addressing the mechanisms of metformin and TRPγ in promoting lipolysis under starvation are interrupted by the lipid storage data in the R2 cells in the current Figure 5A-5E. In addition, presenting Figure 5A-5E before or together with Figure 7 will help readers appreciate the expression of Dh44-R2 and its function in regulating lipid metabolism in Figure 7.

We did.

(4) It might be misleading to use the word "sated" for the condition of 5-hour mild starvation. The word "mild starvation" or the equivalents might be a better word choice.

We appreciate the reviewer’s concern. As hemolymph sugar level does not drop down significantly in 5 hr starvation, the previous papers (Dus et al 2015, Dhakal et al 2022) indicated it as sated condition. To use the word consistently, we prefer using “sated” instead of “mild starvation”.

(5) It is unclear what the white arrows are pointing at in Figures 7O and 7P. Some of those seem to be non-specific signals, so it is hard to connect the figure to the conclusion in Lines 351-353. It would be helpful to add some explanations to help readers interpret Figures 7O and 7P.

In the previous version, Figure 7O and 7P white arrows represented the expression of Dh44R2 in the SEZ region of the brain and R2 region of the gut. In revised version, to make clear, we performed additional immunostaining for the co-expression of trpγ and Dh44R2 in the gut. We found that trpγ and Dh44R2 co-expressed at the R2 region of the gut specifically (Figure 7O and P). Similarly, we found that two cells of Dh44R2 co-expressed in Dh44 cells in the PI region of the brain (now Figure 7Q). We updated this part. lines 375-380.

(6) The figure legend for the (G) panel in Figure 2-figure Supplement 1 was mislabeled as (F).

We corrected it.

(7) In Line 85, the authors might want to write "… among these mutants, only trpγ mutant displayed reduced carbohydrate levels, suggesting …". Please confirm the information for the sentence. lines 87-88.

We clarified it.

**Reviewer #2 (Recommendations For The Authors):**
(1) The trpγ[G4] would be difficult for non-*Drosophila* researchers to understand; it would be better to use trpγ-Gal4.

We got the mutant line from Dr. Craig Montell who named it. We explained it like the following in the main text: “controlled by *GAL4* knocked into the *trpg* locus (*trpgG4* flies; +)” line 109.

(2) The arrows in Figures 7O and 7P need to be explained in the figure legends.

We did.

**Reviewer #3 (Recommendations For The Authors):**
(11) Lines 95-96 should have a reference.

We did.

(12) Lines 129-130: It should read "TRPγ expressed in DH44 cells is sufficient for the regulation of lipid levels."

We changed it as suggested.

(13) Figure 5E needs to be repeated with more trials.

We increased the n numbers. Previously (Figure 5E) we included area of 10 LDs from 3 samples, and in revised figure (Figure 6I) we have included 28 LDs from 10 samples.

(14) Figures 5F-I, bold lines are not too visible and therefore, dotted lines could be used.

We changed it as suggested.

(15) Line 356: It is not true that D-trehalose or D-fructose is commonly detected by DH44 neurons. These sugars at concentrations much higher than the physiological concentration range stimulate DH44 neurons (see Dus et al., 2015).

We removed it.

(16) Lines 362-363: It should read "Expression of TRPγ in DH44 neurons was necessary and sufficient to regulate the carbohydrate and lipid levels.".

We changed it.

(17) Lines 369-370: The authors need to consider removing the possible role of CRF in regulating lipid homeostasis. It could be considered to be far-fetched.

We removed it.

(18) Line 407-408: the sentence "Nevertheless, it is also known that DH44 neurons mediate the influence of dietary amino acids on promoting food intakes in flies (37)" needs to be removed. They used amino acid concentrations that were far greater than the physiological levels observed in the internal milieu of flies. Still, many laboratories cannot reproduce the result of using the high AA concentrations.

We removed it.